# DISTRIBUTION SHIFT RESILIENT GNN VIA MIXTURE OF ALIGNED EXPERTS

## ABSTRACT

The ability of Graph Neural Networks (GNNs) to generalize to diverse and unseen distributions holds paramount importance for real-world applications. However, previous works mostly focus on addressing specific types of distribution shifts, *e.g.,* larger graph size or node degree, or inferring distribution shifts from data environments, which is highly limited when confronted with nuanced distribution shifts. For example, a node in a social graph may have both increased interactions and features alternation, while its neighbor nodes may see different shifts. Failing to consider such complex distribution shifts will largely hinder the generalization effect in practice. Here we introduce GraphMETRO, a novel framework based on a mixture-of-experts (MoE) architecture, enhancing GNN generalizability for both node- and graph-level tasks. The core concept of GraphMETRO includes the construction of a hierarchical architecture composed of a gating model and multiple expert models that are aligned in a common representation space. Specifically, the gating model identifies the significant mixture components that govern the distribution shift on a node or graph instance. Each aligned expert produces representations invariant to a type of mixture component. Finally, GraphMETRO aggregates the representations from multiple experts to produce an invariant representation *w.r.t.* the complex distribution shift for the prediction task. Moreover, GraphMETRO provides interpretations on the distribution shift type via the gating model and offers insights into real-world distribution shifts. Through the systematic experiments, we validate the effectiveness of GraphMETRO which outperforms Empirical Risk Minimization (ERM) by 4.6% averagely on synthetic distribution shifts and achieves state-of-the-art performances on four real-world datasets from GOOD benchmark, including a 67% and 4.2% relative improvement over the best previous method on WebKB and Twitch datasets.

## 1 INTRODUCTION

While Graph Neural Networks (GNNs) (Hamilton et al., 2017; Kipf & Welling, 2017; Dwivedi et al., 2023) excel when trained on data from specific domains (*i.e.,* source data), their ability to generalize to unseen data (*i.e.,* target data) remains a critical concern (Knyazev et al., 2019; Zhang et al., 2017). This challenge can be attributed to the intricate complexities of real-world graph data, which can substantially deviate from the distribution of source data. Within this framework, the ability to manage data from diverse distribution shifts emerges as a fundamental requisite for applications that depend on GNNs, *e.g.,* social networks (Berger-Wolf & Saia, 2006; Greene et al., 2010) and recommendation systems (Ying et al., 2018), where unforeseen variations and novel graph structures are common.

A line of previous research has focused on addressing specific types of distribution shifts. For example, works have looked at distribution shifts related to graph size (Bevilacqua et al., 2021; Buffelli et al., 2022; Yehudai et al., 2021), feature noise (Knyazev et al., 2019; Ding et al., 2021), and node degree or local structure (Wu et al., 2022b; Gui et al., 2022), assuming that the target datasets adhere to the corresponding type of distribution shift. However, these approaches are highly limited as the distribution shift types could be multiple in the real-world datasets and may not be consistent with the presumed distribution shifts. While previous invariant learning methods on graphs (Wu et al., 2022c;a; Sui et al., 2023; Chen et al., 2022) is able to accommodate multiple distribution shifts inferred from data environments, they focus on common patterns within each environment

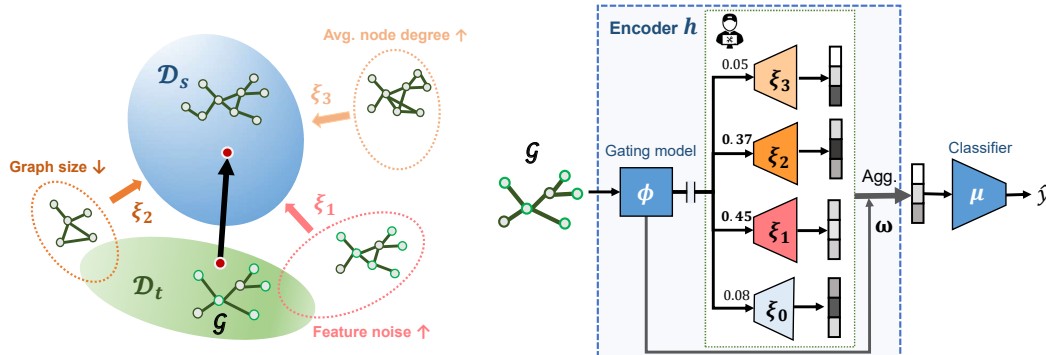

(a) **High-level concept of GraphMETRO.**    (b) **Model architecture of GraphMETRO.**

Figure 1: (a) This example shows three mixture components that controls graph distribution shifts, *i.e.,* feature noise, smaller graph size, and higher node degree. GraphMETRO's goal is to generate invariant representation for the graph instance *w.r.t.* its exhibited distribution shifts. (b) GraphMETRO employs a gating model to identify the significant mixture components that govern the distribution shift on an instance and each expert produces representations invariant to a type of mixture component. The resulting aggregated representation is then used for the prediction task.

and do not explicitly model the variety across node or graph instances. Consequently, models may still fail to generalize when confronted with the alternative and nuanced distribution shifts.

The fundamental challenge is due to the multifaceted property of graph distribution shifts (Gui et al., 2022), which may involve changes of individual node features, alterations in the local structures, and transformations affecting the global or local pattern of the graph. For example, in a citation network, both the feature-level shifts and structure-level changes can contribute to the target distribution shift. And each type of the shift introduces its own unique set of complexities that require to be effectively modeled. Moreover, it is crucial to recognize that the shifts can vary significantly among different instances. For example, one user in a social graph might experience a decrease in interactions, while another user might see a trend of engaging with a certain topic instead. These nuanced distribution shifts play a vital role in accurately characterizing the dynamics of graph data.

As shown in Figure 1, we present a novel framework, GraphMETRO, to enhance model generalizability on both node- and graph-level tasks conditional on each instance. GraphMETRO decomposes any distribution shift into several mixture components, each characterized by its own statistical properties. For example, in Figure 1a, the distribution shift on a graph instance is decomposed into two mixture components controlling feature noise and graph size, respectively. We leverage a graph extrapolation technique to construct these mixture components. Consequently, we break down the generalization goal into two parts: (1) Inferring the distribution shift for an instance based on the mixture components, instead of relying on environmental patterns, and (2) Addressing the distribution shift by mitigating the shifts influenced by individual mixture components.

We design a hierarchical architecture composed of a gating model and multiple expert models, inspired by the mixture-of-experts (MoE) architecture (Jordan & Jacobs, 1994). Specifically, as shown in Figure 1b, the gating model processes an input graph to pinpoint the critical mixture components that govern the given graph instance. Each expert model excels in generating invariant representations with respect to one kind of mixture component, while all of the experts are aligned in a common representation space to ensure model compatibility. Subsequently, GraphMETRO combines representations based on the weights and aligned expert outputs, yielding the final invariant representations that are utilized for predictive tasks.

This process effectively generates invariant representation across multiple types of distribution shifts corresponding to the mixture components, enhancing generalization and the model's ability to make more reliable predictions. To highlight, our method achieves the best performances on four real-world datasets from GOOD benckmark (Gui et al., 2022), including both node and graph classification datasets, where GraphMETRO exhibits a 67% relative improvement over the best baseline on WebKB dataset (Pei et al., 2020). On synthetic datasets, our method outperforms Empirical Risk

Minimization (ERM) by 4.6% in average. Additionally, the gating model outputs the weights over the mixture components indicating the distribution shifts posed on the node or graph instance, which offers interpretations and insights into distribution shifts of unknown datasets.

The key benefits of GraphMETRO are as follows

- It provides a simple yet novel paradigm, which formulates graph generalization as inferring the equivalent mixture as a proxy which is tackled to mitigate the distribution shifts.

- Through the proposed training framework, it effectively mitigates complex distribution shifts which involves multiple and nuanced distribution shifts. and greatly improves GNN generalizability, achieving the state-of-the-art performance on both node and graph classification tasks.

- It provides interpretations about which distribution types occur in the data, offering insights into the intricate nuances of real graph distributions.

## 2 RELATED WORKS

**GNN Generalization**. The prevailing invariant learning approaches assume that there exist an underlying graph structure (*i.e.,* subgraph) (Wu et al., 2022c; Li et al., 2022b;a; Yang et al., 2022; Sui et al., 2022) or representation (Arjovsky et al., 2019; Wu et al., 2022a; Chen et al., 2022; Bevilacqua et al., 2021; Zhang et al., 2022) that is invariant to different environments and / or causally related to the label of a given instance. For example, Yang et al. (2022) explore molecule representation learning in out-of-distribution (OOD) scenarios by directing the molecule encoder to utilize stable and environment-invariant substructures relevant to the labels, and Sui et al. (2022) introduce causal attention modules to identify key invariant subgraph features that can be described as causing the graph label. Besides, Ma et al. (2021) is a theoretical work which studies GNN generalization and examine model fairness, showing that the test subgroup's distance from the training set impacts GNN performance. However, this line of research focuses on group patterns without explicitly considering nuanced (instance-wise) distribution shifts, making its applicability limited. Similar to models on other data modalities, GNN demonstrates resilience to data perturbations which incorporates augmented views of graph data (Ding et al., 2022). Previous works have explored augmentation *w.r.t.* graph sizes (Zhu et al., 2021; Buffelli et al., 2022; Zhou et al., 2022), local structures (Liu et al., 2022), and feature metrics (Feng et al., 2020). Recently, Jin et al. (2023) proposed to adapt testing graphs to graphs with preferably similar pattern as the training graphs. Although these techniques enhance out-of-distribution performance, they may lead to a degradation in in-distribution performance because of the GNN's limited capacity to encode a broader distribution. Another line of research uses attention mechanism to enhance generalization. For example, GSAT (Miao et al., 2022) injects stochasticity to the attention weights to block label-irrelevant information. However, Knyazev et al. (2019) shows that attention mechanism helps GNNs generalize only when the attention is close to optimal. It is also worth mentioning that graph domain adaptation (Zhang et al., 2019; Wu et al., 2020), different from the problem studied in this work, commonly relies on limited labeled samples from target datasets for improved transferability. For instance, to generate domain adaptive network embedding, DANE (Zhang et al., 2019) uses shared weight graph convolutional networks and adversarial learning regularization, while UDA-GCN (Wu et al., 2020) employs attention mechanisms to merge global and local consistencies. Among all, our method introduces a new class, which is built on top of an equivalent mixture for graph generalization to capture multiple and nuanced distribution shifts. While previous graph generalization methods mostly focus on either node- or graph-level task, GraphMETRO can be applied to both tasks.

**Mixture-of-expert models**. The applications on mixture-of-expert models (Jordan & Jacobs, 1994; Shazeer et al., 2017) has largely focused on their efficiency and scalability (Fedus et al., 2022b;a; Riquelme et al., 2021; Du et al., 2022), with a highlight on the image and language domains. For image domain generalization, Li et al. (2023) focuses on neural architecture design and integrates expert models with vision transformers to capture correlations on the training dataset that may benefit generalization, where an expert is responsible for a group of similar visual attributes. For the graph domain, differently motivated as our work, Wang et al. (2023) consider the experts as information aggregation models with varying hop sizes to capture different range of message passing, which aims to improve model expressiveness on large-scale data. GraphMETRO is the first to design a mixture-of-expert model specifically tailored to address graph distribution shifts, coupled with a novel objective for producing invariant representations.

## 3 METHOD

**Problem formulation**. For simplicity, we consider a graph classification task and later extend the application domain to general graph tasks. Consider a source distribution $\mathcal{D}_s$ and an unknown target distribution $\mathcal{D}_t$, we aim to learn a model $f_\theta$ using $\mathcal{D}_s$ such that the model can achieve good task performance in the target distribution. The standard approach is Empirical Risk Minimization (ERM) , *i.e.,*

$$\theta^* = \arg\min_\theta \mathbb{E}_{(\mathcal{G},y)\sim\mathcal{D}_s}\mathcal{L}\left(f_\theta\left(\mathcal{G}\right),y\right), \tag{1}$$

where $\mathcal{L}$ denotes the loss function and $y$ is the label of the graph $\mathcal{G}$. It minimizes the average loss among all examples in the source domain. The underlying assumption of ERM is that data from source and target distribution are independently and randomly sampled from the same underlying distribution (IID). However, the assumption can be easily broken especially for real-world datasets, making $\theta^*$ nonoptimal on the target distribution. Furthermore, an unknown shift in distribution from the source to the target domain precludes the possibility of leveraging supervision during the model training phase, thereby rendering the solution theoretically intractable.

### 3.1 MIXTURE COMPONENTS

To seek a more tractable solution, we propose the following informal assumption:

**Assumption 1 (An Equivalent Mixture for Distribution Shifts)** *Let the distribution shift between the source $\mathcal{D}_s$ and target $\mathcal{D}_t$ distributions be the result of an unknown intervention in the graph formation mechanism. We assume that the resulting shift in $\mathcal{D}_t$ can be modeled by the selective application of up to $k$ out of $K$ classes of stochastic transformations to each instance in the source distribution $\mathcal{D}_s$ ($k < K$).*

Assumption 1 essentially states that the distribution shifts (whatever they are) can be decomposed into several mixture components of stochastic graph transformations. For example, on a social network dataset, each mixture component can represent different patterns of user behavior or network dynamics shifts. Specifically, one mixture component might correspond to increased user activity, while another could signify a particular trend of interaction within a certain group of users. Such mixture pattern is common and well-studied in the real-world network datasets (Newman, 2003; Leskovec et al., 2005; 2007; Peel et al., 2017).

Thus, the assumption simplifies the problem by enabling the modeling of individual mixture components constituting the shift, as well as their respective contributions to a more intricate distributional shift. Previous works (Krueger et al., 2021; Wu et al., 2022c;a) infer such mixture components implicitly from the source distribution, focusing on the variety across different data groups. However, the graph distribution shifts could involve multiple and heterogeneous shifts, *e.g.,* feature-level and structure-level shifts, making it hard to distill the diverse mixture components from the source data. Note that while this assumption may generally apply in practice, as observed later in the experiments, we discuss scenarios that fall outside the scope of this assumption in Appendix F.

**Graph extrapolation as mixture components**. To construct the mixture components without the constraint, we instead employ a data extrapolation technique based on the source data. In particular, we introduce $K$ independent classes of transform function, including random edge removal (Rong et al., 2020), multihop subgraph sampling, and the addition of Gaussian feature noise, *etc.*. The $i$-th class, governed by the $i$-th mixture component, defines a stochastic transformation $\tau_i$ that transforms an input source graph $\mathcal{G}$ into a potentially distinct output graph $\tau_i(\mathcal{G})$, $i = 1, \ldots, K$. For instance, $\tau_i$ can be defined to randomly remove edges with an edge dropping probability in the domain $[0.3, 0.5]$.

We construct a set of stochastic transformations which covers the common graph distribution shifts (Zhao et al., 2021). See Appendix B for the details. Based on the $K$ classes of transform function, we obtain $K$ extrapolated datasets that depict the effects of the mixture components, which we will use next to find the appropriate representation for the mixture observed in each test instance.

### 3.2 MIXTURE OF ALIGNED EXPERTS

In light of the mixture components, we decompose the generalization issue into two distinct facets: (1) Estimating distribution shift on any instance as a function of the relevant mixture components,

and (2) Addressing the identified distribution shift by mitigating individual mixture components. Inspired by the mixture-of-expert (MoE) architecture (Jordan & Jacobs, 1994), the core idea of GraphMETRO is to build a hierarchical architecture composed of a gating model and multiple expert models, where the gating model identifies the significant mixture components that control the given instance and each expert produces representations invariant to one type of mixture component in a common representation space. Finally, our architecture combines these representations into a final representation, which is enforced by our training objective to be invariant to the stochastic transformations within the mixture distribution.

**The design of gating model**. We introduce a GNN $\phi$, which takes any graph as input and outputs the weights $\boldsymbol{w}$ on the mixture components. These output weights serve as indicators, suggesting the most probable mixture components from which the input graph originates. Thus, we regard the model $\phi$ as the gating model to break down the distribution shift into a mixture of weighted mixture components. For example, in Figure 1b, given an unseen graph with decreased graph size and node feature noise, the gating model should assign large weights to the corresponding mixture components while assigning small weights to the irrelevant ones, including the mixture component that controls average node degree. Note that $\phi$ should be such that $\boldsymbol{w}_i$, the $i$-th component mixture weight, strives to be sensitive to the stochastic transformation $\tau_i$ but insensitive to the application of other stochastic transformations $\tau_j$, $j \neq i$. This way, determining whether the $i$-th component mixture is present does not depend on other components.

**The design of expert models**. We aim to build $K$ expert models each of which corresponds to a mixture component. Formally, we denote an expert model as $\xi_i : \mathcal{G} \to \mathbb{R}^v$, where $v$ is the hidden dimension and we use $\mathbf{z}_i = \xi_i(\mathcal{G})$ to denote the output representation. An expert model should essentially produce invariant representations (Pan et al., 2011) *w.r.t.* the distribution shift controlled by the corresponding mixture component. However, it is difficult to make every expert an independent function without aligning the expert outputs in a common representation space. Specifically, each expert model may learn its own unique representation space, which could be incompatible with those of other experts or result in loss of information. Moreover, aggregating representations in separate spaces results in a mixture representation space with high variance. The classifier which takes the combined representation as input, such as multi-layer perceptrons (MLPs), may struggle to effectively capture the complex interactions and dependencies among these diverse representations. Thus, aligning the representation spaces of experts is necessary for ensuring compatibility and facilitating stable model training.

To align the experts properly, we introduce the concept of referential invariant representation:

**Definition 1 (Referential Invariant Representation)** *Let $\mathcal{G}$ be an input graph and let $\tau$ be an arbitrary stochastic transform function, with domain and co-domain in the space of graphs. Let $\xi_0$ be a model that encoders a graph into a representation. A referential invariant representation w.r.t. the given $\tau$ is denoted as $\xi^*(\mathcal{G})$, where $\xi^*$ is a function that maps the original data $\mathcal{G}$ to a high-dimensional representation $\xi^*(\mathcal{G})$ such that $\xi_0(\mathcal{G}) = \xi^*(\tau(\mathcal{G}))$ holds for every $\mathcal{G} \in \text{supp}(\mathcal{D}_s)$, where $\text{supp}(\mathcal{D}_s)$ denotes the support of $\mathcal{D}_s$. And we refer to $\xi_0$ as a reference model.*

Thus, the representation space of the reference model serves as an intermediate to align different experts, while each expert $\xi_i$ has its own ability to produce invariant representations *w.r.t.* a stochastic transform function $\tau_i$, $i = 1, \ldots, K$. We include the reference model as a special "in-distribution" expert model on the source data.

Further, we propose two architecture designs for the expert models. A straightforward way is to construct $(K + 1)$ GNN encoders which ensures the expressiveness when modeling the invariant representations. However, this may increase the memory requirement on the computing resource to approximately $(K + 1)$ times that required for training a single model. An alternative approach to reduce memory usage involves constructing a shared module, *e.g.,* a GNN encoder, among the expert models, coupled with a specialized module, *e.g.,* an MLP, for each expert. This configuration can largely reduce memory usage.

**The MoE architecture**. Given an instance, the gating model assign weights $\boldsymbol{w} \in \mathbb{R}^{K+1}$ over the expert models, indicating the distinct shift of the instance. The output weights being conditional on the input instance enables the depiction of a complex target distribution shifts, where these shifts vary across instances. Based on the inferred mixture components, we obtain the outputs of the expert models which eliminates the effect of the corresponding distribution shifts. Then we compute the

final representation via aggregating the representations based on the gating outputs, *i.e.,*

$$h(\mathcal{G}) = \text{Aggregate}(\{(\phi(\mathcal{G})_i, \xi_i(\mathcal{G})) \mid i = 0, 1, \dots, K\})$$

where $h$ is the encoder of $f$. The aggregation function can be a weighted sum over the expert outputs or a selection function that selects the output of the expert with maximum weight, *e.g.,*

$$h(\mathcal{G}) = \sum_{i=0}^{K} \text{Softmax}(\phi(\mathcal{G}))_i \cdot \xi_i(\mathcal{G}) = \text{Softmax}(\boldsymbol{w}) \cdot [\mathbf{z}_0, \dots, \mathbf{z}_K]^T \tag{2}$$

Consider the distribution shift is controlled by only one mixture component, *i.e.,* $k = 1$, which is signified by the gating model, we can obtain $h(\tau_i(\mathcal{G})) = \xi_i(\tau_i(\mathcal{G})) = \xi_i(\mathcal{G}) = h(\mathcal{G})$ for $i = 0, \dots, K$. This indicates that $h$ automatically produces invariant representations *w.r.t.* any one mixture component out of the $K$ mixture components, while the mixture component can still differ across difference instances. For clarity, we define $\tau^{(k)}$ as a joint stochastic transform function composed of any $k$ or less transform functions out of the $K$ transform functions. We refer to the scenario where $h$ produces invariant representations *w.r.t.* $\tau^{(k)}$ as $\tau^{(k)}$-invariance. To extend $k$ to higher order ($k > 1$), we design objective in Section 3.3 which enforces $h$ to satisfy to $\tau^{(k)}$-invariance. Thus, the representations used for the prediction task are invariant to $\tau^{(k)}$, which further guarantees model generalization under the scope of the distribution shifts covered by $\tau^{(k)}$. Finally, a classifier $\mu$, *e.g.,* an MLP, takes the aggregated representation for the prediction task, and we have $f = \mu \circ h$.

### 3.3 TRAINING OBJECTIVE

As shown in Figure 1b, we consider three trainable modules, *i.e.,* the gating model $\phi$, the experts models $\{\xi_i\}_{i=0}^{K}$, and the classifier $\mu$. Note that the encoder $h$ defined in Equation 2 includes the gating model and the experts. Overall, we optimize them via

$$\min_{\theta} \mathcal{L}_f = \min_{\theta} \mathbb{E}_{(\mathcal{G},y)\sim\mathcal{D}_s} \mathbb{E}_{\tau^{(k)}} \{ \text{BCE}(\phi(\tau^{(k)}(\mathcal{G})), Y(\tau^{(k)})) +$$

$$\text{CE}(\mu(h(\tau^{(k)}(\mathcal{G})), y)) + \lambda \cdot d(h(\tau^{(k)}(\mathcal{G})), \xi_0(\mathcal{G})) \} \tag{3}$$

where $Y(\tau^{(k)}) \in \{0,1\}^{K+1}$ is the ground truth vector, and its $i$-th element is 1 if and only if $\tau_i$ composes $\tau^{(k)}$. BCE and CE are the Binary Cross Entropy and Cross Entropy function, respectively. $d(\cdot, \cdot)$ is a distance function between two representations, $\lambda$ is a parameter controlling the strength of distance penalty. In the experiments, we use Frobenius norm as the distance function, *i.e.,* $d(\mathbf{z}_1, \mathbf{z}_2) = \frac{1}{n}\|\mathbf{z}_1 - \mathbf{z}_2\|_F = \frac{1}{n}\sqrt{\sum_{i=1}^{n}(\mathbf{z}_{1i} - \mathbf{z}_{2i})^2}$, and use $\lambda = 1$ for all the experiments.

Specifically, the gating model $\phi$ is optimized via the first loss term, which aims to improve its accuracy in predicting the distribution shift types. Since the task of the gating model is to identify the significant mixture components, we set apart the other loss terms from backpropagating to it to avoid interference with the training of the gating model. The second loss term aims to improve the accuracy of the encoder output in predicting the graph class. For the third term $d(h(\tau^{(k)}(\mathcal{G})), \xi_0(\mathcal{G}))$, when $\tau^{(k)}$ is only composed of one type of transform function, it aligns the representation spaces of $\xi_i$ ($i > 1$) with the reference model $\xi_0$, which fulfills the condition of referential invariant representations. When $\tau^{(k)}$ is composed of multiple types of transform functions, it enforces $h$ to be $\tau^{(k)}$-invariant, *i.e.,* the output representations are invariant *w.r.t.* the stochastic transform function.

We optimize the objective via stochastic gradient descent, where $\tau^{(k)}$ is sampled at each gradient step. Therefore, our GraphMETRO framework yields a MoE model, which comprises a gating model with high predictive accuracy, and expert models that are aligned and can generate invariant representations in a shared representation space, and a task-specific classifier that utilizes robust and invariant representations for class prediction.

### 3.4 DISCUSSION AND ANALYSIS

**Node classification tasks**. While GraphMETRO mainly focuses on graph classification, it is readily adaptable for node classification. Instead of generating graph-level features, GraphMETRO can produce node-specific invariants. We use transform functions on a graph and identify the distribution shift for each node as mixture components, which is consistent with the objective in Equation 3.

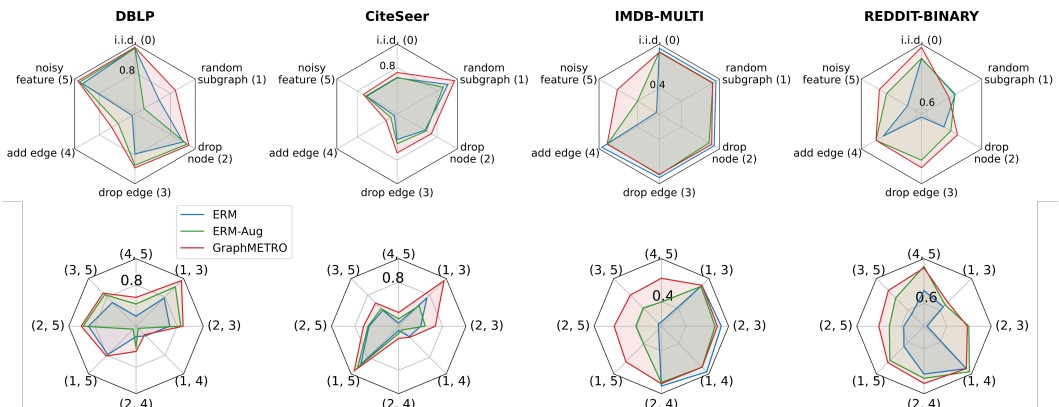

Figure 2: **Accuracy on synthetic distribution shifts**. The first row is the testing accuracy on distribution shifts created from single tranformations. We label the distribution by the clockwise order. The second row is the testing accuracy on distribution shifts created from compositional tranformations, where each testing distribution is a composition of two different transformations. For example, *(1, 5)* denotes a testing distribution where each graph is transformed from the composition of *random subgraph (1)* and *noisy feature (5)* transformations. See Appendix E for the numerical results.

**Intepretability**. The gating model of GraphMETRO predicts the mixture components indicating the distribution shifts posed on the node or graph instance. This offers interpretations and insights into distributions shifts of unknown datasets. In contrast, the prevailing research on GNN generalization (Wu et al., 2022c; Miao et al., 2022; Chen et al., 2022; Wu et al., 2022a) often lacks proper identification and analysis of distribution shifts prevalent in real-world datasets. This missing piece results in a gap between human understanding on the graph distribution shifts and the actual distribution dynamics. To fill the gap, we provide an in-depth study in the experiments to show the insights of GraphMETRO into the intricate nuances of real graph distributions.

**Computational cost**. Consider the scenario where we use an individual encoder for each expert. The forward process of $f$ involves $O(K)$ forward times using the weighted sum aggregation (or $O(1)$ if using the maximum selection). Since we extend the dataset to $(K + 1)$ times larger than the original data, the computation complexity is $O(K^2|\mathcal{D}_s|)$, where $|\mathcal{D}_s|$ is the size of source dataset.

## 4 EXPERIMENTS

In this section, we perform comprehensive experiments on both synthetic (Section 4.1) and real-world datasets (Section 4.2) to validate the generalizability of GraphMETRO across diverse distribution shifts. Subsequently, we extract insights from the underlying mechanisms of GraphMETRO in Section 4.4 and demonstrate how our method interprets distribution shifts in real-world datasets.

### 4.1 INVESTIGATING GRAPHMETRO VIA SYNTHETIC EXPERIMENTS

We initiate our study via a synthetic study to validate the effectiveness of our method.

**Datasets**. We use graph datasets from citation and social networks. For node classification tasks, we use DBLP (Fu et al., 2020) and CiteSeer (Yang et al., 2016). For graph classification tasks, we use REDDIT-BINARY and IMDB-MULTI (Morris et al., 2020). We include the dataset processing and details of the transform functions in Appendix A due to space limitation.

**Training and evaluation**. We adopt the same encoder architecture for Empirical Risk Minimization (ERM), ERM with data augmentation (ERM-Aug), and the expert models of GraphMETRO. For the training of ERM-Aug, we augment the training datasets using the same transform functions we used to construct the testing environments. Finally, we select the model based on the in-distribution validation accuracy and report the testing accuracy on each environment from five trials. See Appendix A for the detailed settings and hyperparameters.

Figure 2 illustrates our model's performance across single (the first row) and compositional transformations (the second row). In most test scenarios, GraphMETRO exhibits significant improvements or performs on par with two other methods. Notably, on the IMDB-MULTI dataset with noisy node features, GraphMETRO outperforms ERM-Aug by 5.9%, and it enhances performance on DBLP

| | Node classification | | Graph classification | | Require domain information |
|---|---|---|---|---|---|
| | WebKB | Twitch | Twitter | SST2 | |
| ERM | $14.29 \pm 3.24$ | $48.95 \pm 3.19$ | $56.44 \pm 0.45$ | $80.52 \pm 1.13$ | No |
| DANN | $15.08 \pm 0.37$ | $48.98 \pm 3.22$ | $55.38 \pm 2.29$ | $80.53 \pm 1.40$ | No |
| IRM | $13.49 \pm 0.75$ | $47.21 \pm 0.98$ | $55.09 \pm 2.17$ | $80.75 \pm 1.17$ | Yes |
| VREx | $14.29 \pm 3.24$ | $48.99 \pm 3.20$ | $55.98 \pm 1.92$ | $80.20 \pm 1.39$ | Yes |
| GroupDRO | $17.20 \pm 0.76$ | $47.20 \pm 0.44$ | $56.65 \pm 1.72$ | $81.67 \pm 0.45$ | Yes |
| Deep Coral | $13.76 \pm 1.30$ | $49.64 \pm 2.44$ | $55.16 \pm 0.23$ | $78.94 \pm 1.22$ | Yes |
| SRGNN | $13.23 \pm 2.93$ | $47.30 \pm 1.43$ | NA | NA | Yes |
| EERM | $24.61 \pm 4.86$ | $51.34 \pm 1.41$ | NA | NA | No |
| DIR | NA | NA | $55.68 \pm 2.21$ | $81.55 \pm 1.06$ | No |
| GSAT | NA | NA | $56.40 \pm 1.76$ | $81.49 \pm 0.76$ | No |
| CIGA | NA | NA | $55.70 \pm 1.39$ | $80.44 \pm 1.24$ | No |
| GraphMETRO | $\mathbf{41.11 \pm 7.47}$ | $\mathbf{53.50 \pm 2.42}$ | $\mathbf{57.24 \pm 2.56}$ | $\mathbf{81.87 \pm 0.22}$ | No |

Table 1: **Test results on the real-world datasets.** We use ROC-AUC as the evaluation metric on Twitch dataset and Accuracy on the others. Each result of GraphMETRO is repeated five times.

by 4.4% when dealing with random subgraph sampling. In some instances, GraphMETRO even demonstrates improved results on ID datasets, such as a 2.9% and 2.0% boost on Reddit-BINARY and DBLP, respectively. This might be attributed to slight distribution shifts in the randomly split testing datasets or the increased model width enabled by the MoE architecture, enhancing the expressiveness on the tasks.

## 4.2    APPLYING GRAPHMETRO TO REAL-WORLD DATASETS

Following our synthetic study, we proceeded to perform experiments on real-world datasets, which introduced more complex and natural distribution shifts. In these scenarios, the testing distribution might not precisely align with the mixture mechanism encountered during training.

**Datasets**. We use four classification datasets, *i.e.,* WebKB (Pei et al., 2020), Twitch (Rozemberczki & Sarkar, 2020), Twitter (Yuan et al., 2023), and GraphSST2 (Yuan et al., 2023; Socher et al., 2013) with the same train-val-test split from the GOOD benchmark (Gui et al., 2022), which exhibit various real-world distribution shifts.

**Baselines**. We use ERM and domain generalization baselines including DANN (Ganin et al., 2016), IRM (Arjovsky et al., 2019), VREx (Krueger et al., 2021), GroupDRO (Sagawa et al., 2019), Deep Coral (Sun & Saenko, 2016). Moreover, we compare GraphMETRO with robustness / generalization techniques developed for GNNs, including DIR (Wu et al., 2022c), GSAT (Miao et al., 2022) and CIGA Chen et al. (2022) for graph classification tasks, and SR-GCN (Zhu et al., 2021) and EERM (Wu et al., 2022a) for node classification task.

**Training and evaluation**. We summarize the architectures and optimizer in Appendix A. Also, we use an individual GNN encoder as the expert architecture for the experiments in the main paper and include the results of designing a shared module among experts in Appendix C, due to space limitation. For evaluation metrics, we use ROC-AUC on Twitch and classification accuracy on the other datasets following Gui et al. (2022).

In Table 1, we observe that GraphMETRO consistently outperforms the baseline models across all datasets. It achieves remarkable improvements of 67.0% and 4.2% relative to EERM on the WebKB and Twitch datasets, respectively. When applied to graph classification tasks, GraphMETRO shows notable improvements, as the baseline methods exhibit similar performance levels. Importantly, GraphMETRO can be applied to both node- and graph-level tasks, whereas many graph-specific methods designed for generalization are limited to one of these tasks. Additionally, GraphMETRO does not require any domain-specific information during training, *e.g.,* the group labels on training instances, distinguishing it from methods like SRGNN.

The observation that GraphMETRO being the best-performing method demonstrates its significance for real-world applications since it excels in handling unseen and wide-ranging distribution shifts. This adaptability is crucial as real-world graph data often exhibit unpredictable shifts that can impact model performance. Thus, GraphMETRO' versatility ensures its reliability across diverse domains,

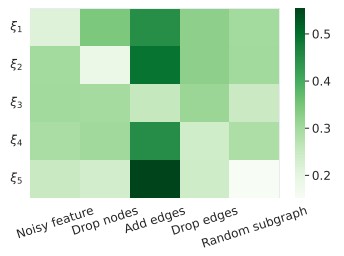

(a) **An example of invariance matrix**

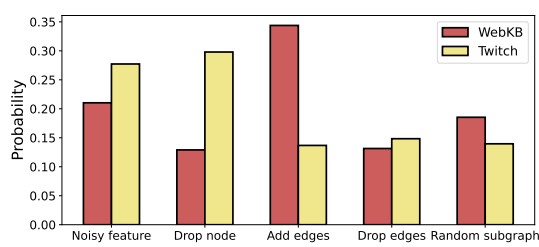

(b) **Probability distribution on the testing distribution shifts**

Figure 3: (a) Invariance matrix on the Twitter dataset. Smaller values (lighter colors) indicate higher invariance of representation produced by each expert *w.r.t.* the corresponding stochastic transform function. (b) Probability distribution predicted by the gating function averaged over the testing instances. Higher probability indicates a potential strong component on the testing distribution.

safeguarding performance in complex real-world scenarios. In Appendix D, we also provide a study about the impact of the stochastic transform function choices on the model performance to analyze the sensitivity and success of GraphMETRO.

### 4.3 INVARIANCE MATRIX FOR INSPECTING GRAPHMETRO

A key insight from GraphMETRO is that each expert excels in generating invariant representations concerning a stochastic transform function. To delve into the modeling mechanism, we define an invariance matrix denoted as $I \in \mathbb{R}^{K \times K}$. This matrix quantifies the sensitivity of expert $\xi_i$ to the stochastic transform function $\tau_j$. Specifically, for $i \in [K]$ and $j \in [K]$, we have

$$I_{ij} = \mathbb{E}_{\mathcal{G} \sim \mathcal{D}_S} \mathbb{E}_{\tau_j} [d(\xi_i(\tau_j(\mathcal{G})), \xi_0(\mathcal{G}))]$$

Ideally, for a given transform function, the representation produced by the corresponding expert should be most similar to the representation produced by the reference model. Therefore, we anticipate the the diagonal entries $I_{ii}$ to be smaller than the off-diagonal entries $I_{ij}$ for $j \neq i$ and $i = 1, \ldots, K$. In Figure 3a, we visualize the normalized invariance matrix computed for the Twitter dataset, revealing a pattern that aligns with our expectations. This demonstrates how GraphMETRO effectively adapts to various distribution shifts, indicating that our approach generates consistent invariant representations for specific transformations through the experts' contributions.

### 4.4 DISTRIBUTION SHIFT DISCOVERY

After obtaining the trained MoE model, we aim to understand the distribution shifts in the testing data. Using the gating model's output weights, each corresponding to a distinct human-interpretable shift, we investigate unseen graph mixtures. We conducted specific case studies on the WebKB and Twitch datasets due to their substantial performance improvements. Specifically, we trained the gating model for multitask binary classification with $(K + 1)$ classes, achieving high accuracies of 92.4% on WebKB and 93.8% on Twitch datasets. When testing with an unknown shift, we computed the gating function's average outputs, revealing global probability distributions indicating shifts. On WebKB, increased edges dominate, while Twitch shows shifts in user language-based node features and fewer nodes. These align with dataset structures: WebKB's diverse university domains and Twitch's language-based user segmentation. Quantitatively validating these observations in complex graph distributions remains a challenge. Future work aims to explore these complexities, offering insights into shifts influenced by temporal or physical dynamics in graph datasets.

## 5 CONCLUSION

This work focuses on the application of graph generalization approaches for diverse, unknown, and inherently complex distribution shifts. We regard graph distribution shifts, by nature, as a mixture of components, where each component has its unique complexity to control the direction of shifts. And we introduce a novel mixture of aligned experts to solve the distribution shift challenge, coupled with an objective to ensure the resulting aggregated representations remain invariant. Our experiments demonstrate significant performance improvements of our method (*ours*) across synthetic and real-world datasets, showcasing its ability to identify different distribution. For future works, we discuss detailed directions in Appendix F, including extending GraphMETRO to domain adaptation settings.

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

# A    EXPERIMENTAL DETAILS

**Open-source code claim**. All of the codes including dataset processing procedures, model construction, and training pipeline will be made public.

**Experimental settings on synthetic datasets**. We randomly split the original dataset into training (80%), validation (20%), and testing (20%) subsets. We consider the transformations when $k = 2$, *i.e.,* $\tau^{(2)}$, which includes the single types of the transform functions and the composition of two different transform functions. For the compositions, we exclude the trivial combination, *i.e.,* adding edges and dropping edges, and the combination that is likely to render empty graph, *i.e.,* random subgraph sampling and dropping nodes. Then, we apply the transform functions on the testing datasets to create multiple variants as the testing environments.

**Model architecture and optimization**. We summarize the model architecture and hyperparameters on synthetic experiments (Section 4.1) in Table 2. We use an Adam optimizer with wight decay 0.0. The encoder (backbone) architecture including number of layers and hidden dimension are searched based on the validation performance on an ERM model, and then fixed for each encoder during the training of GraphMETRO.

|  | Node classification | | Graph classification | |
|---|---|---|---|---|
|  | DBLP | CiteSeer | IMDB-MULTI | REDDIT-BINARY |
| Backbone | Graph Attention Networks (GAT) (Veličković et al., 2018) | | | |
| Activation | PeLU | | | |
| Dropout | 0.0 | | | |
| Number of layers | 3 | 3 | 2 | 2 |
| Hidden dimension | 64 | 32 | 128 | 128 |
| Global pool | NA | NA | global add pool | global add pool |
| Epoch | 100 | 200 | 100 | 100 |
| Batch size | NA | NA | 32 | 32 |
| ERM Learning rate | 1e-3 | 1e-3 | 1e-4 | 1e-3 |
| GraphMETRO Learning rate | 1e-3 | 1e-3 | 1e-4 | 1e-3 |

Table 2: Architecture and hyperparameters on synthetic experiments

For the real-world datasets, we adopt the same encoder and classifier from the implementation of GOOD benchmark[1]. Results of the baseline methods except for Twitter (which is recently added to the benchmark) are reported by the GOOD benchmark. We summarize the architecture and hyperparameters we used as follows

|  | Node classification | | Graph classification | |
|---|---|---|---|---|
|  | WebKB | Twitch | Twitter | SST2 |
| Backbone | Graph Convolutional Network (Kipf & Welling, 2017) | | Graph Isomorphism Network (Xu et al., 2019) w/ Virtual node (Gilmer et al., 2017) | |
| Activation | ReLU | | | |
| Dropout | 0.5 | | | |
| Number of layers | 3 | | | |
| Hidden dimension | 300 | | | |
| Global pool | NA | NA | global mean pool | global mean pool |
| Epoch | 100 | 100 | 200 | 200 |
| Batch size | NA | NA | 32 | 32 |
| ERM Learning rate | 1e-3 | 1e-3 | 1e-3 | 1e-3 |
| GraphMETRO Learning rate | 1e-2 | 1e-2 | 1e-3 | 1e-3 |

Table 3: Architecture and hyperparameters on real-world datasets

---

[1] https://github.com/divelab/GOOD/tree/GOODv1

For all of the datasets, we conduct grid search for the learning rates of GraphMETRO due to its different architecture compared to traditional GNN models, where GraphMETRO has multiple GNN encoders served as the expert modules.

## B    STOCHASTIC TRANSFORM FUNCTIONS

We built a library consists of 11 stochastic transform functions on top of PyG[2], and we used 5 of them in our synthetic experiments for demonstration. Note that each function allows one or more hyperparameters to determine the impact of shifts, *e.g.,* the probability in a Bernoulli distribution of dropping edges, where certain amount of randomness remains in each stochastic transform function.

```
stochastic_transform_dict = {

    'mask_edge_feat': MaskEdgeFeat(p, fill_value),
    'noisy_edge_feat': NoisyEdgeFeat(p),
    'edge_feat_shift': EdgeFeatShift(p),
    'mask_node_feat': MaskNodeFeat(p, fill_value),
    'noisy_node_feat': NoisyNodeFeat(p),
    'node_feat_shift': NodeFeatShift(p),
    'add_edge': AddEdge(p),
    'drop_edge': DropEdge(p),
    'drop_node': DropNode(p),
    'drop_path': DropPath(p),
    'random_subgraph': RandomSubgraph(k)

}
```

We also note that there is an impact on the model performance with different sets or numbers of transform functions. Specifically, we use stochastic transform functions as the basis of the decomposed target distribution shifts. Ideally, the transform functions should be diverse and covers different potential aspects of distribution shifts. However, using a large number of transform functions poses higher expressiveness demand on the gating model, which is required to distinguish different transformed graphs. Moreover, it could also result in an increasing computational costs as the parameter size increases with the number of experts or base transform functions. We include an ablation study in Appendix D to further validate the analysis.

In the practice, we found that the stochastic transform functions works effectively on the real-world datasets, which might indicate their representativeness on the distribution shifts. We believe it would be intriguing to further explore the common base transform functions in the real-world shift in the aid to reconstruct a complex distribution shift.

## C    DESIGN CHOICES OF THE EXPERT MODELS

|  | WebKB | Twitch | Twitter | SST2 |
|---|---|---|---|---|
| GraphMETRO | 41.11 | 53.50 | 57.24 | 81.87 |
| GraphMETRO (Shared module) | 29.05 | 52.77 | 57.15 | 81.71 |

Table 4: Experiment results on comparing different design choices of the expert models.

In the main paper, we discussed the design choices in expert models, highlighting the potential trade-off between model expressiveness and memory utilization. In this section, we delve deeper into various design options and their impact on model performance. Specifically, we investigate a configuration where multiple experts share a GNN encoder while possessing individual MLPs for customizing their output representations derived from the shared module. Our findings and comparative results are presented in Table 4.

---

[2]https://github.com/pyg-team/pytorch_geometric

Notably, our experiments reveal a decrease in model performance. We attribute these performance declines to a potential limitation in the expressiveness of the customized module. This limitation may hinder the module's ability to align with the reference model while simultaneously ensuring that the experts remain invariant to their respective mixture components. This phenomenon draws parallels with data augmentation approaches, as "being invariant to every distribution shifts" using one module may be insufficient. Nevertheless, employing a shared module for the experts continues to yield superior results compared to the baseline models in Table 1. These improvements can be attributed to two key factors: firstly, the selective mechanism of the gating model, which effectively identifies and employs more relevant experts to address distribution shifts; secondly, our designed objective function, which guarantees the generation of invariant representations.

## D STUDY ON THE CHOICE OF TRANSFORM FUNCTIONS

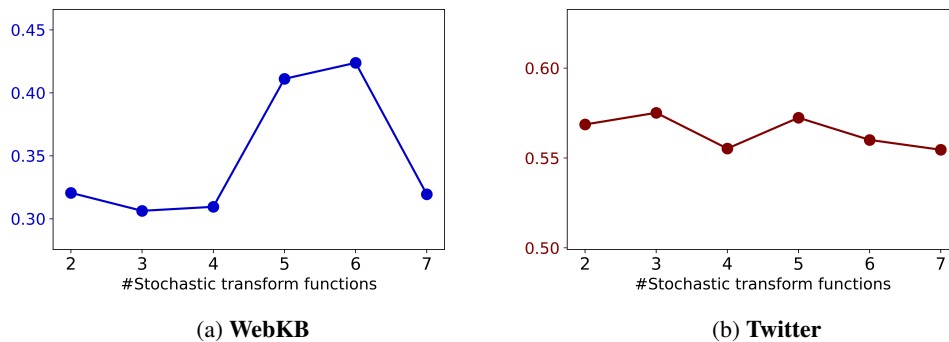

(a) **WebKB**  (b) **Twitter**

Figure 4: The impact of transform function choices on model performance. Note that each number of transform functions corresponds to a particular set of transform functions.

We investigate how the choices of stochastic transform functions affects the performance of Graph-METRO, ranging from 2 to 7 functions. These functions are considered sequentially in the following order:

```
[noisy_node_feat, add_edge, drop_edge, drop_node,
random_subgraph, drop_path, node_feat_shift]
```

where we take the first $n$ transform functions and their paired combinations (exclude trivial combinations like adding edge with dropping edges) during the training of GraphMETRO. We do not from considering all combinatorial choices, such as selecting $n$ distinct functions from the available seven, due to computational constraints. Nonetheless, we maintain our interest in exploring different transform function choices for training GraphMETRO .

Figure 4 illustrates the results for the WebKB and Twitter datasets. A consistent trend emerges: as the number of stochastic transform functions increases, performance tends to decline. For instance, on the WebKB dataset, performance decreases from 42.4% to 31.9%. Similar gradual declines are observed on the Twitter dataset. This phenomenon may be attributed to two factors: (1) Some of the stochastic transform functions may introduce noise that is orthogonal to the target distribution shifts we aim to model, thereby degrading the final aggregated representation. (2) As the number of transform functions grows, the gating function's expressiveness may become insufficient, leading to increased noise and inadequate prediction of the mixture.

## E NUMERICAL RESULTS OF THE ACCURACY ON SYNTHETIC DISTRIBUTION SHIFTS.

In Table 5 and 6, we include the numerical results on the synthetic datasets corresponding to Figure 2 for more precise interpretation. We also compute the average performance across different extrapolated testing datasets, where we see an improvement of

| | DBLP | | | CiteSeer | | |
|---|---|---|---|---|---|---|
| | ERM | ERM-Aug | GraphMETRO | ERM | ERM-Aug | GraphMETRO |
| i.i.d. (0) | 85.71 | 85.66 | 85.92 | 75.80 | 76.00 | 78.01 |
| random subgraph (1) | 84.48 | 85.29 | 85.78 | 75.47 | 75.82 | 77.01 |
| drop node (2) | 71.08 | 74.85 | 76.61 | 62.21 | 63.89 | 66.22 |
| drop edge (3) | 79.69 | 82.34 | 82.95 | 71.48 | 73.24 | 77.00 |
| add edge (4) | 83.41 | 84.44 | 84.98 | 74.29 | 74.87 | 77.26 |
| noisy features (5) | 76.90 | 72.81 | 81.32 | 85.28 | 82.97 | 88.43 |
| (1, 3) | 77.63 | 81.04 | 81.71 | 70.37 | 71.42 | 74.97 |
| (2, 3) | 81.99 | 83.65 | 84.26 | 73.60 | 74.06 | 76.11 |
| (1, 4) | 79.69 | 68.62 | 80.31 | 84.47 | 86.36 | 88.56 |
| (2, 4) | 70.55 | 74.01 | 75.10 | 62.13 | 63.53 | 65.73 |
| (1, 5) | 71.52 | 68.27 | 71.05 | 66.89 | 62.59 | 67.32 |
| (2, 5) | 77.73 | 81.13 | 81.85 | 70.19 | 72.21 | 76.77 |
| (3, 5) | 79.59 | 84.49 | 87.14 | 78.24 | 73.29 | 89.18 |
| (4, 5) | 70.40 | 74.16 | 76.18 | 61.64 | 63.53 | 66.42 |
| Average | 77.88 | 78.63 | 81.08 | 72.29 | 72.41 | 76.36 |

Table 5: Numerical results on synthetic node classification datasets

| | IMDB-MULTI | | | REDDIT-BINARY | | |
|---|---|---|---|---|---|---|
| | ERM | ERM-Aug | GraphMETRO | ERM | ERM-Aug | GraphMETRO |
| i.i.d. (0) | 50.17 | 49.28 | 49.16 | 72.93 | 73.02 | 75.94 |
| random subgraph (1) | 34.30 | 39.94 | 45.86 | 62.59 | 69.03 | 71.22 |
| drop node (2) | 50.42 | 48.73 | 48.83 | 70.01 | 72.27 | 72.26 |
| drop edge (3) | 49.66 | 48.94 | 48.83 | 59.13 | 70.55 | 72.51 |
| add edge (4) | 49.64 | 48.14 | 48.90 | 65.18 | 67.28 | 69.34 |
| noisy features (5) | 50.17 | 49.28 | 49.16 | 68.66 | 68.50 | 66.79 |
| (2, 3) | 34.55 | 40.32 | 45.11 | 58.72 | 64.06 | 66.50 |
| (1, 4) | 34.32 | 40.28 | 46.01 | 59.40 | 62.81 | 65.29 |
| (2, 4) | 34.57 | 40.17 | 46.79 | 61.34 | 66.02 | 66.71 |
| (1, 5) | 49.31 | 48.36 | 48.68 | 65.89 | 66.88 | 68.09 |
| (2, 5) | 50.51 | 48.78 | 48.79 | 68.72 | 69.77 | 68.76 |
| (3, 5) | 49.38 | 47.72 | 48.35 | 55.36 | 65.21 | 64.87 |
| (1, 3) | 48.72 | 48.36 | 48.76 | 61.08 | 61.71 | 62.57 |
| (4, 5) | 34.62 | 39.88 | 46.15 | 62.99 | 68.68 | 68.34 |
| Average | 44.31 | 45.58 | 47.82 | 63.71 | 67.56 | 68.51 |

Table 6: Numerical results on synthetic graph classification datasets

# F  OPEN DISCUSSION AND FUTURE WORKS

**The performance of gating model**. One factor that affect the performance of GraphMETRO is the effectiveness of gating model in identifing distribution shifts from transform functions. Specifically, some transform functions are inherently disentangled, *e.g.,* adding nodes feature noise and random subgraph extraction. In this case, there will be certain distinction between any pair from these three data distributions, *i.e.,* (graphs with node noise, random subgraph graphs, random subgraphs with node noise), which the gating model can easily tell. While some transform functions can be essentially similar, *e.g.,* dropping path and dropping edges, this won't affect the performance of our method as long as each expert outputs the corresponding invariant representation. Lastly, indeed, there could be more complex combinations of the transform functions, which poses challenges to the gating model's expressiveness in identifying the combinations. To improve the gating model's performance, one could initialize it with a model pretrained on a wide variety of data. Since the gating model is required to output the mixture of a node or graph (after it is finetuned on the extrapolated dataset), by enhancing the gating model's predictive capability regarding mixtures, GraphMETRO's final representation should become more resilient. This becomes particularly advantageous when dealing with graphs not previously encountered in the extrapolated dataset.

**In-depth comparison with invariant learning methods**. An interesting view to see the innovation of GraphMETRO is that it breaks the typical invariant learning formulation, which assumes the data is manipulated by the environment variables which are then "decoded" into multiple environments. Instead, GraphMETRO sees the distribution shifts on an instance as a mixture, which is represented

by the score vector output by the gating function over the basis of the transform functions. In other words, GraphMETRO can produce infinite environments as the elements in the score vector are continuous. Once we limit the output domain of the gating function into, *e.g.,* binary $\{0, 1\}$, Graph-METRO can also produce a limited number of environments, *i.e.,* if we categorize the instances based on the score vector, which covers the environment construction in invariant learning. More-over, as mentioned, we propose the concept of referential invariant representation with a base model $\xi_0$, which is also different from previous works on invariant learning.

**The applicability of GraphMETRO**. A key question *w.r.t.* the applicability of GraphMETRO is that, how does the predefined transform functions cover complex distributions causing the distribution shift?

- For general domain, in our experiments, we mainly use the five stochastic transform functions, which are universal graph augmentations as listed in Zhao et al. (2021). In our code implementation, we have also included additional transform functions as shown in Appendix B. These transform functions, while not exhaustive, still cover a wide range of distribution shifts observing from our experimental results. Nevertheless, the real graph distribution shifts can go beyond any possible combinations of the predefined transform functions. In that case, the assumption may not hold, meaning that GraphMETRO may not capture and precisely mitigate the unknown distribution shift. This scenario could always possibly exist due to the lack of information about the testing distribution or its domain knowledge, which is a limitation of our current work.

- However, for specific domains, we can leverage additional knowledge to infer the tendency of the distribution shifts, such as increasing malicious users in a trading system. These information would be very helpful in constructing the transform functions that cover the target distribution shifts well. Specifically, such knowledge can come from two sources: **i) Domain knowledge**, *e.g.,* on molecular datasets, the transform function could be adding additional carbon structure to a molecule (while preserving its functional groups). Or, in a particular social network, transform functions can be defined from known user behaviors. **ii) Leveraging a few samples from target distribution (*i.e.,* domain adaptation).** Specifically, we can leverage the samples from the target distribution to inform the selection or construction of transform functions, which can better guarantee the distribution shifts are covered by the transform functions. For example, we can select more relevant transform functions by, *e.g.,* measuring the distance of the extrapolated datasets under a certain transform function with the target samples in the embedding space. We believe this would be an interesting future direction.

**Label distributional shifts**. In this work, we consider distribution shift in the graph structures and features. We believe applying GraphMETRO to label distributional shift, which is orthogonal and complementary to the focus of our current study, would be an interesting extension. To elaborate, label distributional shifts exert analogous impacts across various modalities, such as graphs or images. Moreover, existing methods Menon et al. (2021); Cao et al. (2019) designed to tackle label distributional shifts can be seamlessly integrated into our proposed framework. Such integration would necessitate minimal adjustments, potentially involving modifications to the loss function or the training pipeline.

