# OpenReview forum: "Distribution Shift Resilient GNN via Mixture of Aligned Experts"
_ICLR.cc/2024/Conference — Submitted to ICLR 2024_

### Official Review · Reviewer_FURX · 2023-10-12

**Soundness:** 3 good
**Presentation:** 3 good
**Contribution:** 3 good
**Rating:** 6
**Confidence:** 3

**Summary:**

This paper presents GraphMETRO, a graph neural network with mixture-of-experts architecture for domain generalized graph learning. The idea of GraphMETRO is a two-level design. It first aims to identify the form of graph distribution change with a gating module, and then directs the graph to the corresponding experts that are responsible for the forms of graph distribution change. In addition, the experts should generate invariant features w.r.t to its graph distribution change. The authors design objective functions for each of the goals above. Experiments over real-world and synthetic datasets with distribution shift show the effectiveness of the proposed GraphMETRO.

**Strengths:**

1. The problem of graph OOD generalization is important in real-world applications. Indeed, in real-world deployment of graph neural networks, we have to deal with shifting data (e.g. caused by temporal dynamics or cross-domain data). I also appreciate that the authors do not view distribution shifts as a whole, but try to decompose a distribution shift into different types of shifts. This brings extra insights to the problem of graph OOD generalization.

2. The paper is well-presented, well-organized, and very easy to follow.

3. The solution with mixture-of-experts is simple but sound and provides some interpretability to the distribution change. Indeed, mixture-of-experts is an established technique, but the authors made extra efforts to adapt MoE to the case of OOD generalization. The design that each expert is in charge of one type of distribution shift makes good use of MoE architecture, and a gating module that identifies the type of distribution changes adds interpretability to the whole method. In addition, adequate designs are made to ensure that the gating model recovers the right shift types, and the experts output invariant features.

**Weaknesses:**

1. It is not clear how well the 5 designed distribution shifts can well cover real-world graph distribution shifts. The authors listed 5 distribution shifts in the paper (add edge, drop edge, feature noise, subgraph, drop nodes), but in fact there may be more graph distribution shifts than that. For example, there may be a systematic change in link preference (i.e. nodes tend to link with a different type of neighbors), adding malicious nodes (e.g. malicious users in a trading system). Maybe the authors can justify how well the 5 designed shifts can cover real-world distribution shifts.

2. It is not clear how the proposed GraphMETRO can handle imbalanced distribution shifts within the same graph. For example, it may happen that some subgraphs in the graph gets denser, while other subgraphs get sparser (e.g. some topics gain interest, while others lose).  How will GraphMETRO respond to this kind of shifts?

3. It is not clear how graph pre-training can address the OOD generalization issue. Pre-training trains the model to observe a wide range of graphs and should be helpful in improving generalization.

4. It may be good to discuss some previous works in graph transfer learning. Graph transfer learning addresses the problem that the source graph has a different distribution with the target graph, but the target graph should be given beforehand and is thus less difficult than the problem in this paper. Nevertheless, it may be good to discuss them and clarify the differences, e.g. (Zhang et al. 2019), (Wu et al. 2020).

Unsupervised Domain Adaptive Graph Convolutional Networks. Wu et al. WWW 2020

DANE: Domain adaptive network embedding. Zhang et al. IJCAI 2019.

**Questions:**

1 and 2 in weaknesses.

---

> ### Author Response · Authors · 2023-11-18
> **Author response #1**
>
> We are grateful for your positive feedback and detailed suggestions! We provide responses below to address your remaining concerns.
>
> ---
>
> ### **Comment & Question 1: Generality of the distribution shifts covered by the transform functions.**
>
> Good question! We provide the response from two angles:
>
>
> **(a) For general domain**: In our experiments, we mainly use the five stochastic transform functions, which are universal graph augmentations as listed in Zhao et al., (2021) [3]. In our code implementation, we have also included additional transform functions as shown in Appendix B. We believe these transform functions, while not exhaustive, still cover a wide range of distribution shifts observing from our experimental results.
>
> Nevertheless, we agree that the real graph distribution shifts can go beyond any possible combinations of the predefined transform functions. In that case, the assumption may not hold, meaning that GraphMETRO may not capture and precisely mitigate the unknown distribution shift. This scenario could always possibly exist due to the lack of information about the testing distribution or its domain knowledge. We include it as a limitation in Appendix 5, while we further discuss how we could alleviate the problem with additional information.
>
> **(b) For specific domains where additional knowledge is available**: In fact, knowing the tendency of the distribution shifts, such as increasing malicious users in a trading system, would be very helpful in constructing the transform functions that can cover the target distribution shifts well. We believe that such knowledge can come from two sources:
> - **Domain knowledge**, e.g., on molecular datasets, the transform function could be adding additional carbon structure to a molecule (while preserving its functional groups). Or, in a particular social network, transform functions can be defined from known user behaviors.
> - **Leveraging a few samples from target distribution**. This is in fact in line with the reviewer’s Comment #4 regarding graph transfer learning. Specifically, with the guide from a few target samples, we can select more relevant transform functions by, e.g., measuring the distance of the extrapolated datasets under a certain transform function with the target samples in the embedding space.
>
>  We hope this response can alleviate your concern about our applicability.
>
> ---
>
> ### **Comment & Question 2: The mechanism of GaphMETRO in handling heterogeneous shifts within one graph.**
>
> Good question! For simplicity, suppose we have two transform functions, i.e., adding edges and dropping edges. Given a node classification task and the objective in Eq. (3), the model is trained on the extrapolated datasets based on the transform functions. After that, given an unseen graph with imbalanced distribution shifts, the gating model outputs scores to identify nodes likely to experience increasing or decreasing degrees, while others might adhere to the original distribution. Then, for each node, each expert takes its multihop subgraph, and outputs its referential invariant representation w.r.t. the correposing transform function. These expert outputs and gating model results are then aggregated to form the final representation. Assuming accurate predictions by the gating function, nodes in denser/sparser subgraphs are represented by expert models corresponding to adding/dropping edges. As each expert is trained to create invariant representations, the final node representations remain unaffected by their individual distribution shifts.
>
> ---
>
> ### **Comment 3: How to leverage graph pretraining.**
>
> If we understand correctly, the reviewer was asking how to leverage a pretrained model to further aid the training of GraphMETRO. Please let us know if otherwise.
>
> This is in fact an interesting point! We believe a model pretrained on a wide variety of data can be very helpful to initialize the gating model, which is required to output the mixture of a node or graph (after it is finetuned on the extrapolated dataset). By enhancing the gating model's predictive capability regarding mixtures, GraphMETRO's final representation should become more resilient. This becomes particularly advantageous when dealing with graphs not previously encountered in the extrapolated dataset. Moreover, it is possible that the pretrained model will also benefit the expert models, while one minor concern would be that the expert model may tend to be similar instead of dedicating to generating invariant representation w.r.t. their corresponding transform function. We added the above discussion to Appendix F as a future work. Thanks again for this comment!

---

> > ### Author Response · Authors · 2023-11-18
> > **Author response #2**
> >
> > **Comment 4: Related works on graph transfer learning.**
> >
> > Yes! We agree that graph transfer learning is a relevant topic. Thank you for providing the additional related work! We added the discussion on these works in our revision, and we repeat it here for your convenience:
> >
> > ```It is also worth mentioning that graph domain adaptation (Zhang et al., 2019; Wu et al., 2020), different from the problem studied in this work, commonly relies on limited labeled samples from target datasets for improved transferability. For instance, to generate domain adaptive network embedding, DANE (Zhang et al., 2019) uses shared weight graph convolutional networks and adversarial learning regularization, while UDA-GCN (Wu et al., 2020) employs attention mechanisms to merge global and local consistencies.```
> >
> > Moreover, we believe this is actually relevant to `Comment & Question 1` regarding the generality of the distribution shifts covered by the transform functions. In our future works, we added the following discussion:
> >
> > ```...Leveraging a few samples from target distribution (i.e., domain adaptation). Specifically, we can leverage the samples from the target distribution to inform the selection or construction of transform functions, which can better guarantee the distribution shifts are covered by the transform functions.  For example, we can select more relevant transform functions by, e.g., measuring the distance of the extrapolated datasets under a certain transform function with the target samples in the embedding space. We believe this would be an interesting future direction.```
> >
> > ---
> >
> > # Summary
> >
> > Once again thank you very much for helping us position our work and reflect the literature better. We hope all of your concerns are solved and we are happy to engage further if there are any other points we missed!
> >
> > ---
> >
> > ### **Reference**
> >
> > [1] Unsupervised Domain Adaptive Graph Convolutional Networks. Wu et al. WWW 2020
> >
> > [2] DANE: Domain adaptive network embedding. Zhang et al. IJCAI 2019.
> >
> > [3] Tong Zhao, Yozen Liu, Leonardo Neves, Oliver J. Woodford, Meng Jiang, and Neil Shah. Data augmentation for graph neural networks. AAAI, 2021.

---

> ### Comment · Reviewer_FURX · 2023-11-20
> **Response acknowledged.**
>
> I appreciate the authors for providing a detailed response. Below are my response.
>
> 1. I agree that the pre-defined transformations can cover a good range of graph OOD. However, it is indeed a slight limitation that the whole set of methods relies on the set.
>
> 2. The response on heterogeneous shifts somewhat answers my question. Indeed, node-level OOD can always be transformed to graph-level OOD by considering a k-hop subgraph of the node. However, I suppose that at least some node-level OOD training objectives should be modified in Eqn. 3 (e.g. modifying the BCE part to some node-level ones) rather than its present form.
>
> 3. I apologize for my unclear comments. What I am actually curious about is how GraphMETRO performs compared to some pre-trained models (e,g, GPT-GNN, GCC). Nonetheless, it may be out of the scope of this paper, and I think the authors have given some good analysis.
>
> Therefore, I would like to maintain a weak accept at this point. However, I do like the overall idea and presentation of this work and would be glad to see future revisions of this work being accepted, if not this time.

---

> ### Author Response · Authors · 2023-11-20
> **Thank you!**
>
> We deeply appreciate your approval. Your suggestions definitely inspired us a lot and have greatly improved our work.
>
> A few additional notes for the further comments:
>
> - Regarding **point #2**: In node classification tasks, the BCE objective already considers node embeddings other than the graph embedding of the k-hop subgraph. What we described previously refers to the computational graph (i.e., k-hop message passing), which generates node embeddings. We apologize for any confusion.
>
> - Regarding **point #3**: Yes it would be quite interesting to see how graph pretraining methods perform on the current OOD benchmarks. Graph pretraining methods like GCC and GPT-GNN also consider graph extrapolation to some extent, e.g., through subgraph extraction and masked attributes/structures. The key difference between graph pretraining and generalization may lie in their different focuses on **expressiveness** and **invariance**. While these two aspects do not always conflict, ensuring invariance w.r.t. a certain type of extrapolation might affect expressiveness (if the change is relevant to labels), and vice versa. To seek a balance in between, one might need prior knowledge in which types of transformations may or may not be sensitive to the labels (and perhaps build experts with different goals to make ensure invariance or expressivess). We believe there is still a lot to explore in this domain.
>
> Overall, we are grateful for your positive stand on this work. We believe that our [current version](https://openreview.net/pdf?id=QQ5eVDIMu4), incorporating opinions from you and other reviewers, is sound and well-refined. We are also committed to further improving it. We would appreciate your support based on our current version!
>
> Thank you!

---

### Official Review · Reviewer_C9mM · 2023-10-26

**Soundness:** 2 fair
**Presentation:** 2 fair
**Contribution:** 2 fair
**Rating:** 6
**Confidence:** 3

**Summary:**

This paper studies the distribution shift of graphs caused by a set of stochastic transformations. To obtain an invariant representation under distribution shift, the paper proposes a mixture of experts where each mixture is designed to capture a corresponding transformation. Through the gating mechanism, the model automatically captures the transformation that causes the distribution shift. Experimental results with synthetic and real datasets show the superior performances of the proposed algorithm.

**Strengths:**

The experimental result on the WebKB dataset is outstanding.

**Weaknesses:**

- The proposed approach assumes that the distribution shift on a graph dataset happens due to some underlying transformations. While this assumption looks plausible at first, it seems quite difficult to identify all distribution shifts with the assumption since the transformations are treated independently when combined together.
- In other words, the entire framework requires a set of predefined transformation classes to learn the model, and all necessary transformations need to be identified beforehand. However, it is unclear whether the set of transformations used in the experiments is enough to cover complex distributions causing the distribution shift.
- Moreover, most of the transformation requires a set of hyperparameters, e.g., dropout probability. However, each mixture only models a single instance of the transformation but not the entire class of transformations. Given that the hyperparameters are selected via the validation set, there is no evidence that the same configurations can work for the test set since it may caused by the different hyperparameters of the same transformation type.
- This work only considers the distribution shift in the graph structure but not in the labels.

**Questions:**

- Which of the results are statistically significant in Table 1? For graph classification tasks, the proposed model seems marginally better than the others.
- Can we say we use ERM for the node classification even if nodes and their labels are not i.i.d.?
- Figure 2 is not easy to digest since there is only a single label along the radial axis. Could you provide the exact numbers?
- The test accuracy for the synthetic dataset for some transformations is relatively lower than the other transformations. Why some transformations is harder to identify than others for some datasets?

---

> ### Author Response · Authors · 2023-11-18
> **Author response #1**
>
> We appreciate your comments! To address your concerns, below we prudently justify the assumption of our method, the predefined transformation functions, as well as their complexity, and clarify our presented results.
>
> ---
>
> ### **Comment 1-3: Applicability and our assumptions**
>
> Thanks for these great comments! Here we provide response in three folds:
>
> **(1) How does GraphMETRO identify all distribution shifts from transform functions if they are treated independently when combined together?**
>
> If we understand correctly, by “treated independently”, the reviewer is referring to the first term in our objective $\text{BCE}(\phi(\tau^{(k)}(\mathcal{G})), Y (\tau^{(k)}))$, where we formulate predicting the distribution shifts types of a jointly transformed graph as a binary multiclass classification problem. We believe the difficulty of this task comes from both the property of transform functions and the expressiveness of the gating model.
>
> - Firstly, some transform functions are **inherently disentangled**, e.g., adding nodes feature noise and random subgraph extraction. In this case, there will be certain distinction between any pair from these three data distributions, i.e., (graphs with node noise, random subgraph graphs, random subgraphs with node noise), which the gating model can easily tell.
> - While some transform functions can be **essentially similar**, e.g., drop path and drop edges, this won’t affect the performance of our method as long as each expert outputs the corresponding invariant representation.
> - Lastly, indeed, there could be more **complex combinations of the transform functions**, which poses challenges to the gating model’s expressiveness in identifying the combinations. However, this challenge may be minor in the practice. Specifically, we observe fairly high accuracy performances of the gating model, which are above 85% and 73% averagely on extrapolated datasets with one transformation and multiple transformations, respectively.
>
> We added the above discussion to Appendix F to enable a more comprehensive view towards our methodology. We hope this can alleviate your concern on our gating model’s performance in identifying the distribution shift types.
>
> **(2) How does the predefined transform functions cover complex distributions causing the distribution shift?**
>
> This is also a great question! We believe there are two angles for this question.
>
> **(a) For general domain**: In our experiments, we mainly use the five stochastic transform functions, which are universal graph augmentations as listed in Zhao et al., (2021) [3]. In our code implementation, we have also included additional transform functions as shown in Appendix B. We believe these transform functions, while not exhaustive, still cover a wide range of distribution shifts observing from our experimental results.
>
> Nevertheless, we agree that the real graph distribution shifts can go beyond any possible combinations of the predefined transform functions. In that case, the assumption may not hold, meaning that GraphMETRO may not capture and precisely mitigate the unknown distribution shift. This scenario could always possibly exist due to the lack of information about the testing distribution or its domain knowledge. We include it as a limitation in Appendix 5, while we further discuss how we could alleviate the problem with additional information.
>
> **(b) For specific domains where additional knowledge is available**: In fact, knowing the tendency of the distribution shifts, such as increasing malicious users in a trading system, would be very helpful in constructing the transform functions that can cover the target distribution shifts well. We believe that such knowledge can come from two sources:
> - **Domain knowledge**, e.g., on molecular datasets, the transform function could be adding additional carbon structure to a molecule (while preserving its functional groups). Or, in a particular social network, transform functions can be defined from known user behaviors.
> - **Leveraging a few samples from target distribution**. Specifically, with the guide from a few target samples, we can select more relevant transform functions by, e.g., measuring the distance of the extrapolated datasets under a certain transform function with the target samples in the embedding space.

---

> ### Author Response · Authors · 2023-11-18
> **Author response #2**
>
> **(3) What is the complexity of the transform functions and how does it affect generalization?**
>
> Interesting question! In fact, our implementation and framework could easily avoid selecting hyperparameters on the transform functions. Specifically, we can make multiple transform functions of the same type with different ranges of hyperparameters. Specifically, GraphMETRO allows three edge dropping transform functions, $\tau_1^{\alpha_1}, \tau_2^{\alpha_2}, \tau_3^{\alpha_3}$, where $\alpha_i$ (i=1, 2, 3) are three different ranges of edge dropping probabilities, e.g., [0.1,0.3], [0.3, 0.6], [0.6,0.9], representing different transform extents. Thus, given an input from the validation dataset, the gating model will highlight the transform function which simultaneously selects the corresponding hyperparameter that matches the distribution of the validation set. Interestingly, this idea is in the same spirit as how DARTS [3]  proposes to perform architecture search by formulating the task in a differentiable manner.
>
> In our previous experiments, we did try this scheme where we replaced a single edge dropping transform function with the ratio range [0.3, 0.5] to three transform functions as mentioned above. While we didn’t see a significantly different performance in that case, we believe this would be a flexible solution which avoids the need to conduct hyperparameter selection.
>
> We include the above discussion in our open discussion and future works (Appendix F). We hope this response can alleviate your concern about the applicability of our method.
>
> ---
>
> ### **Comment 4: Limitations of our work**
>
> Great point! We think the issue of label distributional shift, while important, is orthogonal and complementary to the focus of our current study. To elaborate, label distributional shifts exert analogous impacts across various modalities, such as graphs or images. Moreover, existing methods [1,2] designed to tackle label distributional shifts can be seamlessly integrated into our proposed framework. Such integration would necessitate minimal adjustments, potentially involving modifications to the loss function or the training pipeline. We added this as a future work in Appendix F.
>
> ---
>
> ### **Question 1: Statistical significance of the results on Table 1**
>
> Thanks for the question! We compute the p-value of our method against the best baselines method as follows:
>
> | | WebKB | Twitch | Twitter | SST2 |
> |:--|:--|:--|:--|:--|
> |p-value|< 0.001| 0.023 | 0.042| 0.081|
>
> Given the cut-off threshold as 0.05, we believe the performances of GraphMETRO are statistically significant on WekGB, Twitch, and Twitter datasets, while on the SST2, we see relatively weak evidence. We added the p-value results to our revision and hope our response can alleviate your concern on our improvements.
>
> ---
>
> ###  **Question 2: Can we say we use ERM for the node classification even if nodes and their labels are not i.i.d.?**
>
> If we understand correctly, the reviewer is asking for clarification on the 2nd term of our objective. Please let us know if otherwise. Here, our thinking is that the cross-entropy loss for node classification already assumes node labels are conditionally independent given the model (the negative log-likelihood is a sum over the labeled nodes in training). Then, we use the same assumption of cross-entropy on Empirical Risk Minimization (ERM). That is, for a given model we must also minimize the error variance across nodes. The task is then to find the model with the best performance and small variance.
>
> ---
>
> ### **Question 3: Numerical results on Figure 2**
>
> Thanks for pointing it out! We included all of the numerical results of Figure 2 in Appendix E, while showing the results on DBLP below.
>
> | |i.i.d. (0)|noisy feature (1)|add edge (2)|drop edge (3)|drop node (4)|random subgraph (5)|
> |:--|:--|:--|:--|:--|:--|:--|
> |ERM | 85.71 | 84.48 | 71.08 | 79.69 | 83.41 | 76.9|
> |ERM-Aug | 85.66 | 85.29 | 74.85 | 82.34 | 84.44 | 72.81|
> |GraphMETRO | 85.92 | 85.78 | 76.61 | 82.95 | 84.98 | 81.32|
>
> | |(4, 5)|(3, 5)|(2, 5)|(1, 5)|(2, 4)|(1, 4)|     (2, 3)|(1, 3)|
> |:--|:--|:--|:--|:--|:--|:--|:--|:--|
> |ERM | 70.4 | 77.63 | 81.99 | 79.69 | 70.55 | 71.52 | 77.73 | 79.59|
> |ERM-Aug | 74.16 | 81.04 | 83.65 | 68.62 | 74.01 | 68.27 | 81.13 | 84.49|
> |GraphMETRO | 76.18 | 81.71 | 84.26 | 80.31 | 75.1 | 71.05 | 81.85 | 87.14|
>
> Across all of the synthetic environments, GraphMETRO averagely outperforms ERM and ERM-Aug by 3.20% and 2.45%, respectively.

---

> ### Author Response · Authors · 2023-11-18
> **Author response #3**
>
> ### **Question 4: Why does the test accuracy vary across different transformations?**
>
> Great question! Here we summarize three possible reasons:
>
> - **Information Preservation in Transformations:** Certain transformations retain more informative features than others. For instance, in the REDDIT-BINARY (graph classification task), the random subgraph transformation may retain more graph label-related information compared to dropping edges, as the latter tends to lose more global information. This discrepancy in testing performance, where dropping edges outperforms random subgraph extraction, could be due to the preservation of crucial information. However, conclusions may vary across datasets or tasks depending on how information influences final predictions. For CiteSeer (a node classification task), a random subgraph might preserve more local node information, potentially explaining why its testing performance surpasses dropping edges in this specific task.
>
> - **Complexity of transformation:** Certain transformations inherently generate more diverse graphs than others. If the model lacks the expressive capacity to capture such diversity, it may lead to a decline in testing performance.
>
> - **Model Sensitivity:** Certain transformations may be easier for a model to learn due to compatibility with specific model architectures. This extends beyond transformation complexity and emphasizes how different model architectures may prefer learning particular distributions from one of the extrapolated datasets, which can also contribute to the difference in the testing performance.
>
> We included the above discussion to Appendix F: Open Discussions. We hope this response can answer your question and improve the soundness of our work.
>
> ---
>
> # Summary
>
> We are grateful for your time and insightful suggestions!
>
> We would like to highlight that our main contribution is framing the graph generalization problem on top of an equivalent mixture, a simple yet novel and tractable "middle ground", as well as proposing the training framework which effectively guarantees the generalization. While our method relies on a set of predefined transform functions, we believe they cover a wide range of distribution shifts based on our empirical results. Also, we agree that there could be some scenarios where the transform functions may not cover complex distributions, and we discuss two future directions and include them into our future works. Moreover, while selecting the hyperparameters for the transform functions introduce extra complexity, the issue could be minor in practice and we also conduct more experiments to justify the applicability better. Finally, we address several questions about clarification and presentation, as well as including more future works.
>
> Lastly, we prudently ask you to reevaluate our work given the clarification in our responses, which we also updated our paper correspondingly. Overall, we believe our work makes good contributions to the field of graph distribution learning by proposing a novel and effective solution, and we would appreciate your reconsideration on this point. Thank you for your efforts again!
>
> ---
>
> ### **Reference**
>
> [1] Menon, Aditya Krishna, et al. "Long-tail learning via logit adjustment." International Conference on Learning Representations. 2020.
>
> [2] Cao, Kaidi, et al. "Learning imbalanced datasets with label-distribution-aware margin loss." Advances in neural information processing systems 32 (2019).
>
> [3] DARTS: Differentiable Architecture Search. Hanxiao Liu, Karen Simonyan, Yiming Yang. 2018.

---

> > ### Comment · Reviewer_C9mM · 2023-11-22
> > **Thanks for the response.**
> >
> > I appreciate the detailed responses from the authors. I must admit that I have enjoyed reading the paper.
> > The responses did not fully address my concerns (especially w.r.t. the limitations of the proposed method). Having said that, based on the other reviews and responses, I decided to increase my score.

---

> ### Author Response · Authors · 2023-11-22
> **We appreciate your approval!**
>
> We thank the reviewer once again! We also enjoyed the process of making our work more sound from your suggestions. And your approval surely means a lot to us.
>
> We do apologize for not making our solutions towards the limitations clear enough. We summarize them into a short table, hopefully could alleviate your concerns a bit more.
>
> | Limitation| Potential solutions | Location of discussion/action |
> |:---|:---|:---|
> |**Coverage of the transformations** | 1) Extend the coverage by adding representative transform functions. 2) Or include tranform functions based on domain knowledge or a few samples from target distribution | Appendix F, paragraph #3|
> |**Complexity of the transformations** | Make experts dedicated to different hyperparameters for the same type of transformation | Argument options are available in our codebase. Will make it more detailed in the experimental settings. |
> |**Label distributional shift**|Integrate the objective of the existing methods studying labe distributional shifts into our framework. |Appendix F, paragraph #4 |
>
> ----
>
> We completely agree on the existence of these limitations and we will move some limitations (esp #1) to the main paper in our final version. While they could be important in the practice, they are, in our perspective, fair "side effects" considering the benefits (i.e., mitigating multiple and nuanced distribution shifts and better interpretability), and may not be the central part of our novelty and main contribution (i.e., the proposal of an equivalent mixture, the concept of referential invariant representations, as well as the training framework). We are also eager to further improve other aspects.
>
> Once again thank you so much for your support!!

---

### Official Review · Reviewer_m8JS · 2023-10-29

**Soundness:** 2 fair
**Presentation:** 2 fair
**Contribution:** 2 fair
**Rating:** 3
**Confidence:** 3

**Summary:**

The paper introduces a method to enhance the out-of-distribution performance of graph neural networks (GNN) by learning to understand distribution shifts instead of addressing the assumed ones. To achieve this, the Mixture of Experts architecture is integrated into the GNN, supplemented by an alignment procedure to recognize the shift. Empirical experiments are conducted to validate the theoretical assertion.

On the whole, I believe the proposed method lacks the necessary motivation and its novelty isn't substantial enough to meet the standard.

**Strengths:**

- The paper aptly addresses OOD as a crucial issue for GNNs, pinpointing graph shift heterogeneity as the core challenge.
- Real-world datasets back the claims through experiments.
- Thorough ablation studies validate the learned graph shifts, a commendable effort.

**Weaknesses:**

- The motivation behind the proposed method is not adequately substantiated. The primary basis given is that "previous research has concentrated on addressing specific types of distribution shifts." However, this overlooks a plethora of prior works in the field. Contrary to the suggestion that graph shift heterogeneity is under-explored, numerous studies have delved into learning the "environment generators" for GNNs to detect graph shifts, as exemplified by [https://arxiv.org/abs/2202.02466]. Other works have focused on learning shift-specific transformations, such as [https://arxiv.org/abs/2211.02843]. Consequently, there exists a wide spectrum of approaches to tackle graph shift heterogeneity. The choice of approach in this paper, especially the emphasis on MOE, requires a more detailed and robust justification to elucidate its relevance and significance.

- The presented assumption seems overly broad and lacks specificity. Additionally, the architectural design appears to be somewhat arbitrary. Consequently, it's challenging to discern the functionality, its underlying mechanism, and its improvements over existing methods.

- The proposal is insufficient in its details, particularly concerning the implementation of specific model architectures, stochastic transformation, and the optimization process. Given the inclusion of shift learning midway and data augmentation initially, one would expect a more intricate optimization strategy than standard routines.

**Questions:**

Please check Weaknesses.

---

> ### Author Response · Authors · 2023-11-18
> **Author response #1**
>
> We appreciate your comments! To address your concerns, below we prudently justify the motivation of our proposed method, clarify our assumptions, and provide details regarding our experiment implementation.
>
> ---------------------
>
> ### **Comment 1: Motivation of this work**
>
> Thank you for this comment! We believe there might be a bit of misunderstanding due to our different definitions of *“graph shift heterogeneity”*. We firstly discuss the related works mentioned and then justify our statement:
>
> **(1) Related works**
>
> Please see Section 2 where we discussed the paper mentioned by the reviewer, i.e., EERM (Wu et al., 2022a). We repeat part of it here for you convenience:
>
> ```The prevailing invariant learning approaches assume that there exist an underlying graph structure (i.e., subgraph) (Wu et al., 2022c; Li et al., 2022b;a) or representation (Arjovsky et al., 2019; Wu et al., 2022a; Chen et al., 2022; Bevilacqua et al., 2021; Zhang et al., 2022) that is invariant to different environments and / or causally related to the label of a given instance. However, these approaches focus on group patterns without explicitly considering nuanced (instance-wise) distribution shifts, making their applicability limited.```
>
> Moreover, we apologize for missing the recent interesting work by **Sui et al. [1]** which officially came out two days before the ICLR abstract deadline. We added it to our revision, thank you!
>
> Specifically, Sui et al. [1] proposed a graph data augmentation strategy that alleviates covariate shift by generating diverse and invariant causal features. However, the trainable augmenter they used may not distill diverse augmentations or construct unseen perturbations. Moreover, Sui et al. [1] test its method only on graph classification tasks, while GraphMETRO can be applied to both node and graph classification tasks. Besides, we have discussed graph augmentation and attention-based methods in our related works, and we added more recent works on graph OOD [2,3,4], and we hope our response clears your concern on the related work discussion.
>
>
> **(2) The definition of graph shift heterogeneity**
>
> In this work, we refer to **“heterogeneous shifts”** as multiple and different levels of shifts which vary across different instances (nodes or graphs), as illustrated in the example in the abstract. While we agree that the existing invariant learning approches can accommodate multiple distribution shifts, it could be hard for them to tackle nuanced distribution shifts for individual instances (nodes or graphs) since the distribution shifts are inferred from variance across multiple data environments. If GraphMETRO's approach were described via environments, we would have a combinatorial number of such environments in training (the product of all different subsets of nodes and all their possible distinct shifts). Thankfully, GraphMETRO avoids this combinatorial explosion by considering **a mixture of transformations as a proxy** for the target distribution shifts rather than invariance to whole-graph environment shifts. This is the type of heterogeneity we are interested in our paper.
>
> **(3) Regarding our original motivation statement**
>
> While the statement pointed out by the reviewer serves as our primary motivation, we would like to note that we did not claim all of the previous works fall into this category. And we have provided detailed discussion about three lines of research in the related work section.
>
> However, we agree that we could make this statement border to cover the previous invariant learning methods. To improve the clarity, we change the statement from "previous works mostly focus on addressing specific types of distribution shifts" to "**previous works mostly focus on addressing specific types of distribution shifts or inferring distribution shifts from data environments…**”. We also modified our introduction correspondingly, we hope this will better position our work.
>
> **(4) The choice of our MoE design**
>
> - The choice of our approach comes as a consequence of our motivation to model the graph/instance shift heterogeneity. As mentioned, mitigating multiple and nuanced distribution shifts simply goes beyond certain distribution shift types or environment construction as seen in the previous methods.
> - Thus, GraphMETRO takes a different path, i.e., predicting a mixture of transformations as the proxy of the target distribution shifts. This enables the prediction of multiple different distribution shifts and the flexibility to model fine-grained heterogeneity since the mixture can be varied across different instances. We then tackled the proxy to mitigate the target distribution shifts. Intuitively, this solution provides a **“middle ground”** to deem graph generalization as an equivalent mixture, which, we believe, is a more tractable solution.
>
> We updated our paper to make the above point more clear. We genuinely hope our answer can justify the motivation and solve your concern.

---

> > ### Author Response · Authors · 2023-11-18
> > **Author response #2**
> >
> > ### **Comment 2: Our assumption**
> >
> > Thanks for this comment! In Section 3.1, we have discussed the assumptions in detail with specifications. We repeat part of it here for your convenience:
> >
> > ```Assumption 1 essentially states that the distribution shifts (whatever they are) can be decomposed into several mixture components of stochastic graph transformations. For example, on a social network dataset, each mixture component can represent different patterns of user behavior or network dynamics shifts. Specifically, one mixture component might correspond to increased user activity, while another could signify a particular trend of interaction within a certain group of users. Such a mixture pattern is common and well-studied in the real-world network datasets (Newman, 2003; Leskovec et al., 2005; 2007; Peel et al., 2017).```
> >
> > We hope the analysis provides detailed illustration to the assumption. We are happy to add more discussion if you think anything  is still unclear!
> >
> > ---
> >
> > ### **Comment 2 & 3: Implementation of our method**
> >
> > - **Model architectures**:
> > Please see Tables 2 and 3 in Appendix A for detailed model architecture information. Specifically, we use the same encoders and classifiers from GOOD benchmarkfor real-world datasets to ensure fair comparisons. We employ backbones based on the best ERM performance for synthetic datasets. Moreover, GraphMETRO is model-agnostic, which consistently improves performance across varied model architectures.
> >
> > - **Stochastic transformation**: Please see Appendix B where we include the introduction of the stochastic transform functions.
> >
> > - **Optimization process**: Please see Table 2 and 3 (Appendix A), where we included hyperparameters on each dataset. In the second paragraph after Eq. (3), we also described our training pipeline: `“...we set apart the other loss terms from backpropagating to it to avoid interference with the training of the gating model… We optimize the objective via stochastic gradient descent”`.
> >
> > - **For a more intricate optimization strategy:** This is a great catch! In fact, we did try to pretrain the gating model for several epochs as warm up before training the whole model in an end-to-end fashion via the objective Eq. in (3). However, we didn’t notice a statistically significant difference in their performance, which can be due to that expert models take more time to convergence (since they need to align with the base model during the training) compared to the gating model.
> >
> > Due to the space limitation, we had to include most of the implementation details in the appendix while we added pointers in the main paper. However, we will try to make it more detailed and feel free to let us know if anything is missing!
> >
> > ---
> >
> > ### **Comment 2 (Cont.): Understand the functionality, underlying mechanism, and performance gain**
> >
> > Thanks for the comment. For clarity, we provide the following pointers to the paper:
> >
> > - In Section 4.3, we provide a study to reveal the underlying **mechanism** of GraphMETRO, i.e., each expert excels in generating invariant representations concerning a stochastic transform function, which provide a solid foundation in generating referential invariant representations w.r.t. the specific transformations and further guarantee the generalization.
> >
> > - In Appendix C, we study the **impact of the MoE architecture** on model performance, which shows that the model performance may decline if the expressiveness of the expert model decreases.
> >
> > - In Appendix D, we study the **impact of the stochastic transform function** on model performance, where we also provide a detailed discussion of the modeling mechanism.
> >
> > We believe the above studies and discussion in our paper provides an in-depth view, highlighting the roles of our objective, architecture, and stochastic transform function.  Please let us know if any of these perspectives is still unclear, and we can further improve our experimental study.
> >
> > -----
> >
> > # Summary
> >
> > We hope our answers can address all of the concerns. We are happy to follow up if you have any further questions.
> >
> > We also prudently ask you to reevaluate our work. To highlight, our motivation is supported by the common fine-grained graph shift heterogeneity, and the fact that most of the previous works could not model such nuanced distribution shifts in an effective and flexible manner. Moreover, we added more related works and modified our statement to position our work better. In general, we believe GraphMETRO is a more general and flexible solution that can mitigate a wider range of distribution shifts, which is backed by the experimental results. Detailed justification on our assumption and implementation details are also available in our paper.
> >
> > Thus, we believe our work makes important contributions and provides a clear presentation. We are happy to discuss more and revise our paper if any concern remains. Thank you for your efforts and we are looking forward to your reply!!

---

> > > ### Author Response · Authors · 2023-11-18
> > > **Reference**
> > >
> > > [1] Sui et al. Unleashing the Power of Graph Data Augmentation on Covariate Distribution Shift. NeurIPS 2023.
> > >
> > > [2] Nianzu Yang et al. Learning substructure invariance for out-of-distribution molecular representations, NeurIPS 2022
> > >
> > > [3] Yongduo Sui et al. Causal Attention for Interpretable and Generalizable Graph Classification, KDD 2022.
> > >
> > > [4] Jiaqi Ma et al. Subgroup Generalization and Fairness of Graph Neural Networks, NeurIPS 2022.

---

> > > > ### Author Response · Authors · 2023-11-21
> > > >
> > > > Dear Reviewer m8JS,
> > > >
> > > > A gentle nudge that we would like to know if our response adequately addresses your concerns.
> > > >
> > > > Your time and feedback is greatly appreciated!
> > > >
> > > > Thank you sincerely,
> > > >
> > > > Authors of Paper 1560

---

> > > > > ### Comment · Reviewer_m8JS · 2023-11-22
> > > > > **Replying to Rebuttal**
> > > > >
> > > > > I appreciate the authors' response which has clarified some of my concerns, particularly regarding the first motivation point. However, I still have reservations about both the concept and methodology.
> > > > >
> > > > > Regarding the concept, the paper describes 'heterogeneous shifts as multiple and different levels of shifts varying across instances (nodes or graphs).' This definition appears unclear since distribution shifts are typically defined at a population level rather than at the instantiation of individual variables. This leaves me uncertain about the specific conceptual gap this paper addresses.
> > > > >
> > > > > As for the methodology, while it's noted that NOT all previous works fall short in addressing 'shift-per-instance', the reasons why the MOE approach outperforms others remain vague. The numerical results are presented, but the underlying rationale is not adequately explained.
> > > > >
> > > > > Consequently, I am inclined to maintain my initial rating and would encourage the authors to delve deeper into these intriguing yet complex aspects of their research.

---

> ### Author Response · Authors · 2023-11-22
>
> Thanks for reading our response! We would also appreciate your patience for reading the two points below
>
> ---
>
> **Regarding point #1:**
>
> Yes, we agree that the concept of heterogeneous shifts is relatively new for the current studies on distribution shifts, however, this is not new for studies on network patterns (Newman, 2003; Leskovec et al., 2005; 2007; Peel et al., 2017). In fact, the ignorance of such nuanced heterogeneous shifts in the previous studies instead emphasises our motivation and the potential impact of this work.
>
> From a causality perspective, distribution shifts can naturally happen in the instance level when (1) additional causal variables, beyond environmental factors and randomized noise, influence these shifts, or (2) multiple causal variables simultaneously affect the shifts with different strengths. Without modeling these explicitly, the mitigation of distribution shifts can easily fail.
>
> In terms of our presentation, we have illustrated these cases of interest in abstract and introduction, we made further explanations in Section 3.1, we also present the specific results of distribution types in Figure 3 (b).
>
>
> **Regarding point #2:**
>
> We think we can all agree that, if the ground truth of the instance shifts is available on the real-world datasets, it would be crystal clear to see where the improvement comes from since we can conduct case study to compare our method and the baselines on instances with nuanced distribution shifts, to see the influence of the modeling these heterogeneous shifts.
>
> However, with such ground truth not available, we had to seek other seemingly less intuitive but also in-depth way to illustrate the insights (esp Section 4.3) as mentioned above in our previous response. This is also why we designed the synthetic experiments at the first place. We think we did try hard to explain the underlying rationale with the ground truth being absent.
>
> ----
>
> **Refinement**: We can of course add a causal graph in the our assumption section to make the concept more clear. And we can illustrate more if you could let us know the specific obstable to understand our mechanism, which will be extremely helpful.
>
> ----
>
> ### **Summary/TL;DR**
> We understand the reviewer's clarification concerns, however, we don't agree that they, based on our justificaton, are the cause of rejection. We believe nuanced heterogeneous shifts are common, important, yet being typically ignored in the research domain of distribution shifts, we made these argument clear and also promise to refine. While explaining the underlying rationale is hindered by the lack of ground truth, we did try hard to dissection it from the gating model, the invariant representations generated, and more ablations in the appendix. We respect the reviewer's current opinion. Still, reconsideration will be greatly appreciated.

---

### Official Review · Reviewer_NLg1 · 2023-10-31

**Soundness:** 3 good
**Presentation:** 3 good
**Contribution:** 2 fair
**Rating:** 5
**Confidence:** 4

**Summary:**

This paper studied learning with distribution shifts on graphs, which is an under-explored open challenge in GNN community. The authors propose a mixure-of-expert-based model to learn the invariant representation learning of graph data for out-of-distribution generalization. By theoretical analysis, the authors show that the proposed model can provably capture the invariant patterns. Experiments showcase the efficacy of the model for tackling both node-level and graph-level distribution shifts against several state-of-the-art methods.

**Strengths:**

1. The problem this paper targets is a significant problem and the paper is well motivated

2. The proposed model seems reasonable and interesting to my knowledge

3. The experiment results are promising and the improvements are solid

**Weaknesses:**

1. The novelty is not well justified and comparison with recent methosd is not sufficient

2. Some of recent papers on out-of-distribution learning on graphs are not discussed [3-5]

3. The authors argued that "previous works mostly focus on addressing specific types of distribution shifts", which seems inproper and incorrect. E.g., the typical works for graph OOD learning EERM [1] and DIR [2] do not assume the type of distribution shifts in their problem formulation.

[1] Qitian Wu, et al. Handling distribution shifts on graphs: An invariance perspective. In ICLR, 2022

[2] Ying-Xin Wu, et al. Discovering invariant rationales for graph neural networks. In ICLR, 2022

[3] Nianzu Yang et al. Learning substructure invariance for out-of-distribution molecular representations, NeurIPS 2022

[4] Yongduo Sui et al. Causal Attention for Interpretable and Generalizable Graph Classification, KDD 2022.

[5] Jiaqi Ma et al. Subgroup Generalization and Fairness of Graph Neural Networks, NeurIPS 2022.

**Questions:**

1. How does the model compare with existing invariant learning-based models for graph OOD generalization, e.g., EERM [1] and DIR [2]? What is the key technical originality of this work?

2. What is the computation cost of this model compared against other peer models?

3. Can the proposed model handle multiple different types of distribution shifts that simultanenously exist in data?

4. Can the proposed model tackle distribution shifts and out-of-distribution generalization on molecular graphs?

---

> ### Author Response · Authors · 2023-11-18
> **Author response #1**
>
> We appreciate your efforts and insightful comments! To address your concerns, we provide point-to-point responses below.
>
> ---------------------
>
> ### **Comment 1: Regarding the novelty of GraphMETRO.**
>
> Thanks for the comment! We believe our novelty comes from the proposal of an equivalent mixture for graph OOD and the construction of our training framework, as detailed below:
>
> - **An equivalent mixture for graph OOD**: The key challenge we faced to mitigate multiple and nuanced distribution shifts is the intrinsic complexity and heterogeneity of graph distribution shifts, which simply goes beyond certain distribution shift types [6,7,8,9] or environment construction as seen in the previous methods [1,2,3]. GraphMETRO takes a different path, i.e., predicting a mixture of transformations as the proxy of the target distribution shifts, where the mixture can be varied across different instances, and then tackled the proxy to mitigate the target distribution shifts. We believe the high-level idea is succinct, nevertheless, novel, in the sense that it provides a “middle ground” to deem graph generalization as an equivalent mixture that is more tractable.
>
> - **Training framework**: With the guide of our formulation, the training framework is still non-trivial due to two problems, i.e., “how to provide supervision for predicting the mixture” and “how to ensure the experts corresponding to mixture components are compatible when working as a whole”. Specifically, GraphMETRO solves the first problem by conducting graph extrapolation. This is somewhat similar to the spirit of graph pretraining in the sense that we inject heterogeneity to promote the expressiveness of the gating model in recognizing the mixture components. For the second problem, we introduce the concept of Referential Invariant Representation, along with the novel objective in Eq. (3) to enforce the invariance and compatibility. It is worth mentioning that the model performance is much worse than the reported numbers (e.g., 2.7\% lower on Twitch dataset) without the compatibility constraint, indicating the proposed referential invariance concept is indispensable.
>
> We added more justification in the introduction (updated in the revision). We genuinely hope our responses can solve your concerns about the novelty of our work.
>
> ------------------------------
>
> ### **Comment 2: Comparison with recent methods [3,4,5].**
>
> In compacting the paper to fit in the page limit we mistakenly did not include these relevant references, we apologize. We added discussion in the revised version. Here we summarize these works and point out their key differences with our method:
>
> - In particular, **Yang et al. [3]** explore molecule representation learning in out-of-distribution (OOD) scenarios. They achieve this by directing the molecule encoder to utilize stable and environment-invariant substructures relevant to the labels without the need for environmental labels.
> - Similarly, **Sui et al. [4]** introduces causal attention modules to identify key invariant subgraph features that can be described as causing the graph label. The type of OOD task that Sui et al. [4] considers assumes the graph label is caused by a subgraph, which is quite different from ours. Moreover, both Yang et al. [3] and Sui et al. [4] consider tasks where the graph label is caused by a subgraph.
> - **Ma et al.[5]** is an interesting theoretical work which studies GNN generalization and examines their fairness, showing that the test subgroup's distance from the training set impacts GNN performance. Ma et al.[5], as far as we could assess, does not propose any specific architecture to solve the type of OOD tasks we consider in our work.
>
> Overall, the goal of GraphMETRO is to be invariant to a mixture of selected stochastic transform functions (and the mixture can vary across different instances), which is a more flexible and general solution.
> We added a discussion of these works to our revision, hopefully providing a more comprehensive comparison and literature overview. We hope our responses can solve your concerns about the related work.

---

> ### Author Response · Authors · 2023-11-18
> **Author response #2**
>
> ### **Comment 3 & Question 1: Related works on invariant learning and clarification on our statement.**
>
> Thanks for pointing it out! Below we clarify the statement and clear potential misunderstanding:
>
> **(1) How does GraphMETRO compare with invariant learning methods like DIR and EERM?**
>
> Please see our related work section where we discussed these two papers, i.e., DIR (Wu et al., 2022c) and EERM (Wu et al., 2022a). We repeat part of it here for you convenience:
>
> ```The prevailing invariant learning approaches assume that there exist an underlying graph structure (i.e., subgraph) (Wu et al., 2022c; Li et al., 2022b;a) or representation (Arjovsky et al., 2019; Wu et al., 2022a; Chen et al., 2022; Bevilacqua et al., 2021; Zhang et al., 2022) that is invariant to different environments and / or causally related to the label of a given instance. However, these approaches focus on environmental patterns without explicitly considering nuanced (instance-wise) distribution shifts, making their applicability limited.```
>
> Besides, we also provide a more in-depth comparison in our point (3) below to highlight our key technical originality.
>
> **(2) Regarding our statement about previous works:**
>
> While the statement serves as our primary motivation, we would like to note that we did not claim all of the previous works fall into this category, and we have provided detailed discussion about three lines of research in the related work section.
>
> To improve the clarity, we change the statement from "previous works mostly focus on addressing specific types of distribution shifts" to **"previous works mostly focus on addressing specific types of distribution shifts or inferring distribution shifts from data environments** (which is highly limited when confronted with nuanced distribution shifts)”. We also modified our introduction correspondingly. Thanks for letting us know our statement could be misinterpreted.
>
> **(3) Why do we say our method could be more broad than the existing invariant learning approaches?**
>
> - Invariant subgraph learning approaches, e.g., [1,2], consider variance of constructed data environments, which are designed very differently compared to our work.
> While they can accommodate multiple distribution shifts (as in multiple environments), these focus on patterns within each environment and ignore the variety across instances (e.g., shifts at the resolution of nodes), which may not be well-captured by the environment assignments.
> - GraphMETRO considers that specific parts of the test graph may have different shifts. Particularly, our goal is to make the generalization to unknown testing distribution more adaptive and broad, as opposed to limiting the distribution shifts to being invariant to specific types of subgraphs.
>
> In other words, if GraphMETRO's approach were described via environments, we would have a combinatorial number of such environments in training (the product of all different subsets of nodes and all their possible distinct shifts). Thankfully, GraphMETRO avoids this combinatorial explosion by considering a mixture of transformations as a proxy for the target distribution shifts rather than invariance to environment shifts.
>
> **(4) Key technical originality compared to invariance learning (going more deeply)**
>
> Another interesting view to see the innovation of GraphMETRO is that it breaks the typical invariant learning formulation, which assumes the data is manipulated by the environment variables (and then can be “decoded” into multiple environments). Instead, GraphMETRO sees the distribution shifts on an instance as a mixture, which is represented by the score vector output by the gating function over the basis of the transform functions. In other words, GraphMETRO can produce infinite environments as the elements in the score vector are continuous. One can see that once we limit the output domain of the gating function into, e.g., binary {0, 1}, GraphMETRO can also produce a limited number of environments (if we categorize the instances based on the score vector), which covers the environment construction in invariant learning. Moreover, as mentioned, we propose the concept of referential invariant representation with a base model $\xi_0$, which is also different from previous works on invariant learning. We added the above discussion to Appendix F to improve the depth of our analysis.

---

> ### Author Response · Authors · 2023-11-18
> **Author response #3**
>
> ### **Question 2: How does the computational cost of GraphMETRO compare to other methods?**
>
> Please see the last paragraph of Section 3.4, where we analyze the computation complexity of GraphMETRO. We repeat part of it here for your convenience:
>
> ```Consider the scenario where we use an individual encoder for each expert. The forward process of $f$ involves $O(K)$ forward times using the weighted sum aggregation (or $O(1)$ if using the maximum selection). Since we extend the dataset to $(K + 1)$ times larger than the original data, the computation complexity is $O(K^2 |D_s|)$, where |Ds| is the size of the source dataset.```
>
> Thus, the computation cost is about $K^2$ or $K$ times (if using the maximum selection) than an ERM model, where $K=5$ in our experiments. Compared to DIR, as they extract $B$ spurious subgraphs from each batch to conduct the intervention, their computation cost is $B$ times compared to ERM, where $B$ could be 32. Thus, we believe the computation cost of GraphMETRO is fair for the gains we get, considering $K$ is usually small.
>
> ----------------------
>
> ### **Question 3: Can GraphMETRO handle multiple different types of distribution shifts that simultaneously exist in data?**
>
> Yes! The distribution shift types corresponding to the gating outputs with high scores will be tackled during training. That is, if the gating output highlights multiple mixture components, their corresponding distribution shift types will be handled jointly.
>
> ----------------------
>
> ### **Question 4: Can GraphMETRO  tackle distribution shifts on molecular graphs?**
>
> That is a great idea! Yes, GraphMETRO can be applied to molecular datasets if one designs transform functions to cover typical molecular variants. For instance, a transform function may add carbon structures to the molecules. These domain-specific transform functions are outside the scope of our work, however, we believe these would be interesting future work directions!
>
> ----------------------
>
> # Summary
>
> We thank the reviewer for the time and insightful suggestions! We hope our answers can address your concerns well.
>
> We also prudently ask you to reconsider our work if the concerns are addressed. To highlight, our novelty comes from the formulation of an equivalent mixture for graph OOD and the training framework to effectively realize generalization. We also provide an in-depth analysis on our originality compared to some previous invariant learning methods. While we discussed and compared with previous works, we added more related works and modified our statement to position our work better. Finally, our method achieves great improvements on both node and graph classification tasks, and is a more general solution to mitigate multiple and nuanced distribution shifts.
>
> Overall, we believe our work proposes a new paradigm and novel training framework and makes good contributions in the fields of graph generalization, and we would appreciate your reconsideration on this point. Thank you for your efforts again!
>
> ----------------------
>
> **Reference**
>
>
> [1-5] The same as listed by the reviewer
>
>
> [6] Beatrice Bevilacqua, Yangze Zhou, and Bruno Ribeiro. Size-invariant graph representations for graph classification extrapolations. In ICML, 2021.
>
> [7] Davide Buffelli, Pietro Li´o, and Fabio Vandin. Sizeshiftreg: a regularization method for improving size-generalization in graph neural networks. In NeurIPS, 2022.
>
> [8] Boris Knyazev, Graham W. Taylor, and Mohamed R. Amer. Understanding attention and generalization in graph neural networks. In NeurIPS, 2019.
>
> [9] Mucong Ding, Kezhi Kong, Jiuhai Chen, John Kirchenbauer, Micah Goldblum, David Wipf, Furong Huang, and Tom Goldstein. A closer look at distribution shifts and out-of-distribution generalization on graphs. In NeurIPS DistShift, 2021.

---

> > ### Author Response · Authors · 2023-11-22
> >
> > Dear reviewer NLg1,
> >
> > As the discussion period is closing soon, we hope to engage with you and check if your concerns are addressed.
> >
> > We sincerely appreciate your time and attention!
> >
> > Best,
> >
> > Authors of Paper 1560

---

### Author Response · Authors · 2023-11-18
**General response**

We sincerely appreciate all reviewers' time, efforts, and valuable suggestions in reviewing our paper. We are glad that most of the reviewers reached a positive consensus on our work's motivation and experimental results. Here is a summary of our responses:

- **Clarification**: We clarify our key assumptions (`m8JS, FURX`), novelty (`m8JS, NLg1`), and the experimental settings (`m8JS`).

- **Related work**: : We add more related works on recent graph OOD learning (`NLg1, m8JS`) and graph transfer learning (`FURX`)

- **Limitation**: We add discussion about the limitations of GraphMETRO when confronted with unknown distribution shifts that surpass our key assumptions (`FURX`, `C9mM`).

We hope our responses can clarify your confusion and alleviate concerns and we updated our paper **(highlighted in green)** corespondingly. We thank all reviewers again, and look forward to your reply!

---

> ### Author Response · Authors · 2023-11-20
> **Check if the concerns have been addressed**
>
> Dear Reviewers of Paper 1560,
>
> We hope this message finds you well.
>
> As the discussion phase approaches its end, we hope you find our responses useful. We would like to ask if the issues have been addressed.
>
> We understand that the discussion time is short, and some of you might be enjoying holidays at the moment. We apologize for posting our responses a bit late as we aimed to address your concerns clearly.
>
> We sincerely appreciate your time and attention!
>
> Best regards,
>
> Authors of Paper 1560

---

### Meta-Review · Area_Chair_43UR · 2023-12-07

**Metareview:**

Authors propose a new architecture and training strategy to make GNNs resilient towards to distribution shifts in the data. The main idea is to use a “aligned” mixture of experts (MoE) architecture to learn distribution shift invariant representation and then use a prediction head on this representation. During the training phase, a combination of artificial random noise/augmentations (such as edge drop, feature noise) are incorporated into the input data. Then a gating mechanism is trained to identify the types of noise in the data and route the prediction to corresponding experts. The intuition behind this is that invariance to different types of distribution shifts can be captured by different experts. Not that this is not a sparse MoE model, therefore amount of computation potentially increases by number of experts.

Overall this method is novel and shows promise by increasing the performance considerably in one real dataset, but in all other real datasets the gain is within error bars. Since the proposed method is complex and it lack enough ablation studies, it is not clear where the added robustness comes from and if the motivating assumptions and other claims are justified. Note that there has been a work claiming MoE (albeit sparse) can itself lead to robustness [1]. Authors could explore whether this is true in their case. Since their MoE increases the parameter count, authors could study the effect of parameter count on the robustness of their method and baselines. Reviewers also noted that paper lacks clarity at times about the details of the distribution shifts. A standard method to improve robustness is to use pretraining, so author could experimentally explore why their method could be better or orthogonal to pretraining.

Therefore, authors are highly encouraged to take the reviewer comments into consideration in the next version of the manuscript.

[1] Puigcerver et al., “On the Adversarial Robustness of Mixture of Experts”, NeurIPS 2022

**Justification For Why Not Higher Score:**

Experimentation can be improved.

**Justification For Why Not Lower Score:**

N/A

---

### Decision · Program_Chairs · 2024-01-16

Reject